# Molecular mechanism targeting condensin for chromosome condensation

Menglu Wang[1], Daniel Robertson[1], Juan Zou (ID)[1], Christos Spanos (ID)[1], Juri Rappsilber (ID)[1,2] & Adele L Marston (ID)[1✉]

## Abstract

**Genomes are organised into DNA loops by the Structural Maintenance of Chromosomes (SMC) proteins. SMCs establish functional chromosomal sub-domains for DNA repair, gene expression and chromosome segregation, but how SMC activity is specifically targeted is unclear. Here, we define the molecular mechanism targeting the condensin SMC complex to specific chromosomal regions in budding yeast. A conserved pocket on the condensin HAWK subunit Ycg1 binds to chromosomal receptors carrying a related motif, CR1. In early mitosis, CR1 motifs in receptors Sgo1 and Lrs4 recruit condensin to pericentromeres and rDNA, to facilitate sister kinetochore biorientation and rDNA condensation, respectively. We additionally find that chromosome arm condensation begins as sister kinetochores come under tension, in a manner dependent on the Ycg1 pocket. We propose that multiple CR1-containing proteins recruit condensin to chromosomes and identify several additional candidates based on their sequence. Overall, we uncover the molecular mechanism that targets condensin to functionalise chromosomal domains to achieve accurate chromosome segregation during mitosis.**

**Keywords** Condensin; Shugoshin; Lrs4; Pericentromeres; rDNA
**Subject Categories** Cell Cycle; Chromatin, Transcription & Genomics

See also: Cutts et al

## Introduction

Accurate chromosome segregation protects against aneuploidy and requires precise packaging of the genome. Mitotic chromosome architecture is defined by an interplay between the structural maintenance of chromosomes (SMC) complexes cohesin and condensin, which extrude DNA loops (Yatskevich et al, 2019; Hoencamp and Rowland, 2023). Cohesin has gained the additional property of tethering the sister chromatids together during mitosis to provide the cohesion to resist microtubule-dependent pulling forces

(Yatskevich et al, 2019). In all SMC complexes, the molecular motor is a dimer of SMC proteins whose ATPase heads are bridged by kleisin proteins (Yatskevich et al, 2019). The association of the SMC core complex with a number of accessory proteins modulates their activity and targeting to confer specific functions in looping and/or cohesion. In the case of cohesin and condensin, these accessory proteins are part of a family of hook-shaped HEAT-repeat proteins, termed HAWKs (Wells et al, 2017). Cohesin HAWKs are Scc3/SA2 and the interchangeable Scc2/NIPBL and Pds5 proteins (Yatskevich et al, 2019). Yeast have a single condensin with HAWKs Ycg1 and Ycs4, while vertebrates have condensin I and II, with distinct HAWKs (CAP-G/CAP-D2 or CAP-G2/CAP-D3). In vertebrates, cohesin initiates chromosome separation in interphase (Batty et al, 2023), however, most cohesin is removed during prophase, leaving only the pericentromeric pool holding sister chromatids together (Waizenegger et al, 2000). Compaction during prophase is driven by condensin II which generates long loops that are further divided into shorter nested loops after nuclear envelope breakdown by condensin I (Gibcus et al, 2018). In yeast, cohesin drives the majority of chromosome compaction in mitosis, while condensin has specific roles at the rDNA and pericentromeres (Schalbetter et al, 2017; Lavoie et al, 2004).

At metaphase, mitotic chromosomes attach to microtubules from opposite poles via their sister kinetochores. Kinetochores reside on centromeres, which are embedded within larger chromosomal domains called pericentromeres, whose architecture is defined by cohesin and condensin (Ribeiro et al, 2009; Sacristan et al, 2022; Samoshkin et al, 2009; Sen Gupta et al, 2023; Oliveira et al, 2005). Pericentromeres perform critical structural and regulatory roles in chromosome segregation, including orienting kinetochore-microtubule attachments, resisting spindle forces and engaging surveillance mechanisms to sense improper attachments (McAinsh and Marston, 2022; Hindriksen et al, 2017). Cohesin enrichment at pericentromeres prevents the premature separation of sister centromeres and allows tension establishment (Tanaka et al, 2000; Eckert et al, 2007; Fernius and Marston, 2009; Paldi et al, 2020). Condensin organizes vertebrate centromeres into two domains (Sacristan et al, 2022) and is important in regulating centromere stiffness (Ribeiro et al, 2009). Consistently, condensin protects against merotely (where a single sister kinetochore is captured by microtubules from opposite poles) (Samoshkin et al, 2009).

Initial insights into how SMC proteins are specifically targeted to chromosomal loci to define their architecture came from

[1]Centre for Cell Biology, Institute of Cell Biology, University of Edinburgh, Edinburgh EH9 3BF, United Kingdom. [2]Institute of Biotechnology, Technische Universität Berlin, Gustav-Meyer-Allee 25, 13355 Berlin, Germany. ✉E-mail: adele.marston@ed.ac.uk

budding yeast pericentromeres. Their "point" centromeres are defined by a short ~125 bp core sequence upon which the kinetochore assembles (McAinsh and Marston, 2022). A phosphorylation-dependent interaction between the Ctf19 kinetochore protein and a conserved patch on the Scc4 component of the Scc2-Scc4 cohesin loader targets cohesin to the core centromere (Hinshaw et al, 2017, 2015). Centromere-loaded cohesin extrudes a loop on either side to shape the ~10 kb pericentromere, and convergent genes flanking centromeres set the position of the pericentromere borders by acting as barriers to loop extrusion and sites of sister chromatid cohesion (Paldi et al, 2020). This loop-based structure at pericentromeres provides a platform for surveillance mechanisms that direct and sense the establishment of biorientation (Li et al, 2024). The pericentromeric adapter protein, shugoshin, Sgo1, acts downstream of cohesin and plays a key role in this process. Sgo1 facilitates sister kinetochore biorientation by promoting the pericentromeric recruitment of a number of effectors, including PP2A, aurora B, and condensin. Condensin biases sister kinetochores towards biorientation in early mitosis and promotes the topoisomerase II-dependent removal of residual centromeric catenanes (Verzijlbergen et al, 2014; Baxter et al, 2011). Following the establishment of biorientation, sister kinetochore tension leads to dissociation of Sgo1, condensin, and loop-extruding cohesin from pericentromeres, while cohesive cohesin is retained at pericentromere borders to resist spindle forces (Peplowska et al, 2014; Verzijlbergen et al, 2014; Indjeian et al, 2005).

Specific targeting of SMC complexes through their accessory subunits may be a general principle underlying the architecture of specialized chromosomal domains. The conserved patch on the cohesin loader Scc4 that binds Ctf19 also engages with the RSC chromatin remodeler complex and may contribute to cohesin loading genome-wide (Muñoz et al, 2022). Similarly, the equivalent patch on the Scc4 ortholog, MAU2, mediates cohesin recruitment to chromosomes in Xenopus extracts (Hinshaw et al, 2017), and a small deletion of this region in human MAU2 causes the developmental disorder Cornelia de Lange syndrome (Parenti et al, 2020). Although the molecular basis of condensin targeting is unknown, a number of candidate receptors have been described, in addition to yeast Sgo1 (Verzijlbergen et al, 2014; Peplowska et al, 2014; Yahya et al, 2020). In fission yeast and Candida albicans, monopolin proteins (Lrs4-Csm1/Mde4-Pcs1) promote the centromeric enrichment of condensin to prevent merotely (Tada et al, 2011; Gregan et al, 2007; Burrack et al, 2013). Similarly, monopolin (Lrs4-Csm1) is important for condensin localization and function within the rDNA and mating locus in budding yeast (Dinda et al, 2023; Johzuka and Horiuchi, 2009). In Xenopus egg extracts, the general transcription factor TFIIH promotes the enrichment of condensin on chromatin, though this may be indirect (Haase et al, 2022). In vertebrates, a short sequence in the tail of the chromokinesin Kif4A is implicated in condensin regulation and recruitment to chromatin (Mazumdar et al, 2004; Samejima et al, 2012; Poser et al, 2019; Wang et al, 2020; Takahashi et al, 2016).

Here we reveal that condensin is recruited to budding yeast pericentromeres through a direct interaction between a Short Linear Motif (SLiM) on Sgo1, termed Conserved Region 1 (CR1), and a conserved patch on the HAWK subunit Ycg1. Mutation of specific residues in Sgo1 or Ycg1 that disrupt this interface demonstrates the importance of pericentromeric condensin for pericentromere structure

and efficient sister kinetochore biorientation in mitosis. The same Ycg1 patch, along with a CR1 motif in Lrs4, is also critical for the recruitment of condensin to the rDNA and for rDNA condensation. We further find that chromosomal arm condensation is initiated at metaphase, in response to the establishment of sister kinetochore biorientation, in a manner requiring the Ycg1 patch. Importantly, the Ycg1 patch that recruits condensin to pericentromeres is conserved throughout evolution, including in the human ortholog, CAP-G, and we identify CR1 motifs in other candidate condensin interactors. Our findings suggest a general mechanism by which condensin, and potentially other SMCs, interact with ligands involved in its targeting and regulation.

# Results

## Condensin interacts directly with Sgo1 through its Ycg1 subunit

Condensin subunits co-immunopurified with Sgo1 in metaphase-arrested yeast extracts (Verzijlbergen et al, 2014; Su et al, 2021). We found that immobilized purified full-length V5-Sgo1 on beads specifically pulled down the pentameric condensin holo-complex, indicating Sgo1 interacts with condensin directly (Fig. 1A–C). Sgo1 also caused condensin to elute at a lower volume in analytical SEC (Fig. EV1A). To determine which subunit of condensin Sgo1 interacts with, we performed Crosslinking Mass Spectrometry (CLMS). Mapping of inter-condensin subunit crosslinks onto a previously published cryo-EM structure of tetrameric condensin (lacking Ycg1) (Lee et al, 2020) and a Ycg1 (6–932, Δ499–555)-Brn1(384–529) crystal structure (Kschonsak et al, 2017) (Fig. EV1B; Dataset EV1) confirmed the detection of bonafide interactions. Restricting our analysis to crosslinks involving Sgo1 revealed that the majority of crosslinks mapped to the HAWK subunit Ycg1, with a small number mapping to Smc4 in a region known to be close to Ycg1 (Lee et al, 2020) (Fig. 1D). The largest number of crosslinks were in the Sgo1 C-terminal region, with a smaller number mapping to an N terminal region (Fig. 1D). This suggested that a major Ycg1 binding site resides in the C-terminal region of Sgo1, while a minor binding site may exist in the N terminal region of Sgo1. To address this more directly, we tested whether recombinant Sgo1 N- or C-terminal fragments could associate with condensin from yeast extracts lacking endogenous Sgo1 to preclude multimerization. Condensin binding to Sgo1 fragments immobilized on beads was detected by immunoblotting against Brn1-6HA (Fig. EV1C). Consistent with the CLMS, this indicated that the C-terminal region of Sgo1 (residues 350–590) harbors the predominant condensin binding site but that there is an additional minor binding site in the N-terminal region (residues 1–130). A previous report using peptide arrays suggested that Sgo1 binds condensin through its Serine-Rich Motif (SRM, residues 137–163) (Yahya et al, 2020), however, in our hands, a recombinant fragment (130–190) containing the Sgo1 SRM did not pull down condensin (Fig. EV1C). Sgo1 (350–590) showed the most robust condensin binding and although V5-Sgo1(350–590) itself bound non-specifically to anti-HA beads, immobilized V5-Sgo1(350–590) specifically pulled down pentameric condensin (Fig. EV2A). We conclude that Sgo1(350–590) binds condensin directly through the Ycg1 subunit.

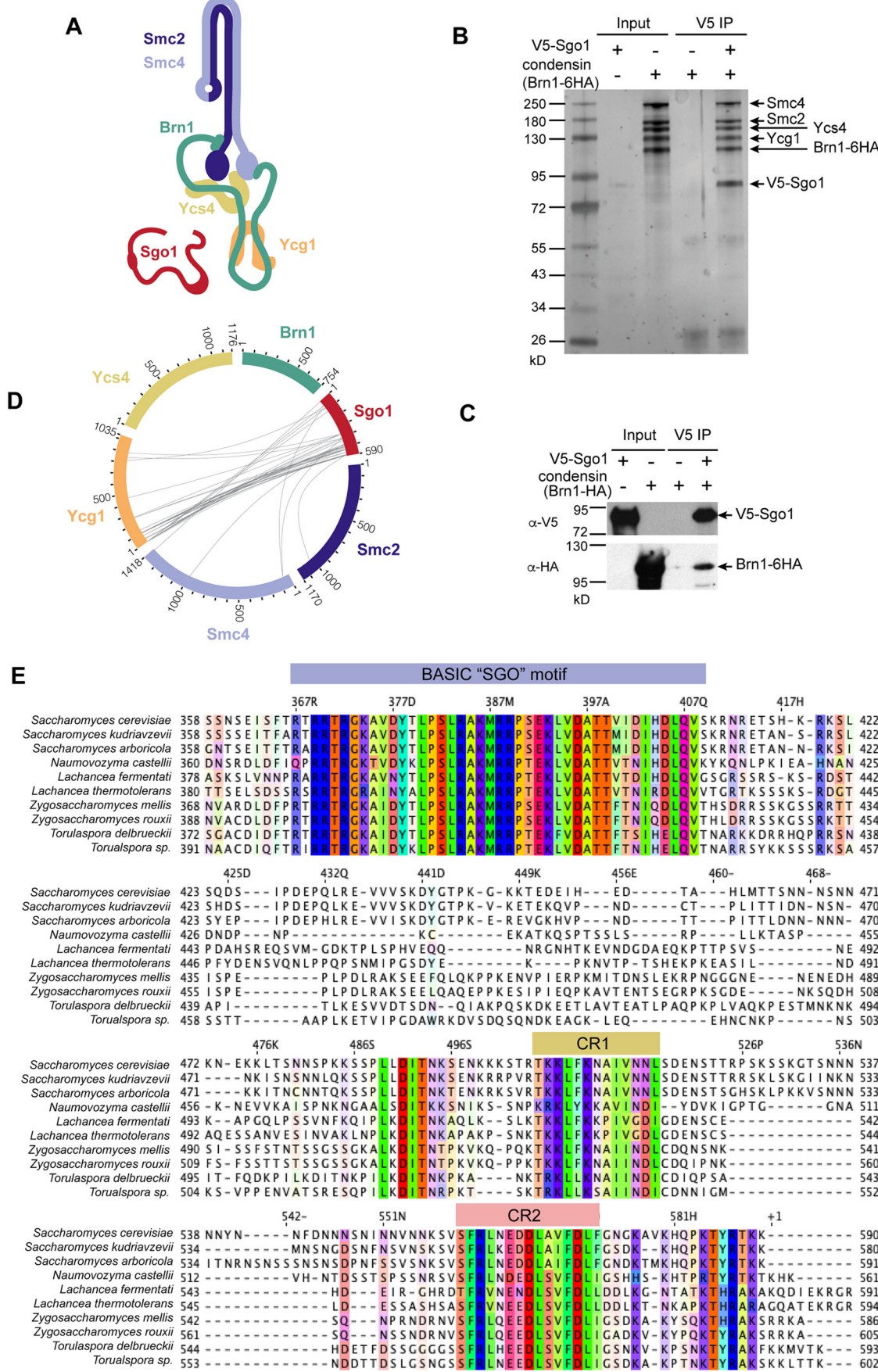

**Figure 1. Identification of a condensin binding motif in Sgo1.**

(A) Schematic representation of *Saccharomyces cerevisiae* condensin complex and Sgo1. (B, C) Recombinant *Sc* Sgo1 co-immunoprecipitated with *Sc* condensin complex using anti-V5-coupled beads. Immunoprecipitates were analyzed by silver-stained SDS-PAGE (B) and immunoblotted with the indicated antibodies (C). Performed once as shown. (D) Crosslinking mass spectrometry of full-length *Sc* Sgo1 and *Sc* condensin holo-complex using crosslinker BS3. Intra-subunit condensin interactions and self-links were hidden, and only crosslinks with scores >10.5 are shown. (E) Sequence alignment of C-terminal region of Sgo1 among related yeast. The basic motif involved in histone binding is highlighted in blue. Two further conserved regions are highlighted in yellow (CR1) and pink (CR2). Source data are available online for this figure.

## CR1—a SLiM in the C-terminal region of Sgo1 mediates Ycg1 binding

To identify the determinants of the Sgo1-Ycg1 interaction we purified Ycg1(6–932, Δ499–555)-Brn1(384–529), as described previously (Kschonsak et al, 2017), and generated a complex with Sgo1(350–590) (Fig. EV2B,C). We performed CLMS and again validated our results by mapping crosslinks onto the reported Ycg1(6–932, Δ499–555)-Brn1(384–529) crystal structure (PBD:5OQQ, (Kschonsak et al, 2017) Fig. EV2D,E; Dataset EV1). Crosslinks between Ycg1(6–932, Δ499–555)-Brn1(384–529) and Sgo1(350–590) confirmed a major interface between the C-terminal region of Sgo1 (residue 470 onwards) and the N terminal region of Ycg1 (Fig. EV2D,E).

Despite significant functional conservation, shugoshins (Sgo) show a high degree of diversity (Marston, 2015), however alignment of the C-terminal region of shugoshins from related yeast species revealed that, in addition to the previously reported basic SGO motif that may associate with histones (Kawashima et al, 2010), there are two further short Conserved Regions (CR1 and CR2) (Fig. 1E). Although Sgo1 is prone to nonspecific binding, direct comparisons among a panel of recombinant fragments focused our attention on the CR1 motif as the most critical determinant for specific binding. We found that recombinant Sgo1 containing the CR1 motif consistently pulled down Ycg1(6–932, Δ499–555)-Brn1(384–529) (Fig. 2A,B). Furthermore, specific deletion of CR1, but not CR2, greatly reduced the ability of Sgo1(350–590) to pull down Ycg1(6–932, Δ499–555)-Brn1(384–529) (Fig. 2C–E). Together, these experiments indicate that a major Ycg1 binding site resides with the CR1 motif of Sgo1.

The Sgo1 CR1 motif is characterized by an arrangement of hydrophobic residues (L508, F509, I513, V514, and L517 in *S. cerevisiae*) flanked by positively charged lysine/arginine residues (Fig. 2F). We mutated the Sgo1 CR1 hydrophobic residues to glutamic acid in various combinations and found that mutation of L508 and F509 had the strongest effect on Ycg1(6–932, Δ499–555)-Brn1(384–529) binding in vitro (Fig. 2G,H). To address the importance of the CR1 motif in vivo, we adapted a tethering assay we developed previously (Verzijlbergen et al, 2014; Nerusheva et al, 2014). We fused full-length Sgo1, or its mutant derivatives, to TetR-GFP in cells carrying *tetO* arrays inserted at a chromosomal arm site (Appendix Fig. S1A). Chromatin immunoprecipitation, followed by qPCR (ChIP-qPCR) showed that, as expected, TetR-GFP-Sgo1 specifically recruited Brn1-6HA to the *tetO* arrays, while mutations within Sgo1-CR1 reduced Brn1 recruitment (Appendix Fig. S1B). Consistent with our in vitro analysis, the Sgo1-L508E F509E mutations had the strongest effect in abolishing Brn1 recruitment to this ectopic site and deletion of the serine-rich motif (SRM) (Sgo1-Δ137–163), previously reported to be important (Yahya et al, 2020), did not prevent Brn1 recruitment (Appendix Fig. S1B). We note that, unexpectedly, combining Sgo1 I513E, V514E, and L517E mutations

with L508E F509E increased Ycg1 binding in vitro (Fig. 2G,H), but since this was not observed in the in vivo tethering assay which uses full-length protein (Appendix S1B), this is likely explained by additional glutamic acid substitutions increasing the nonspecific binding of Sgo1 fragments in vitro. Overall, our data indicate that L508 and F509 glutamic acid substitutions in the context of the full-length protein are sufficient to abrogate condensin association in vivo. Therefore, the Sgo1-Ycg1 interaction depends on the conserved adjacent hydrophobic residues L508 and F509.

## Broad conservation of the predicted binding site in Ycg1/Cnd3/CAP-G condensin proteins

Next, we sought to identify the determinants of Ycg1 that mediate Sgo1 binding. We ran alphafold2 predictions (Evans et al, 2022; Jumper et al, 2021; Mirdita et al, 2022) of Ycg1(6–932, Δ499–555)-Brn1(384–529) and Sgo1(487–522). Out of five highly similar models (Fig. EV3A–C). we considered Rank 5 to be the best predictor of the Sgo1-Ycg1 interface, since it reported the highest pLDDT scores (Mariani et al, 2013) for the two Sgo1 residues L508, F509 we identified to be important experimentally. All five models were consistent with our CLMS of full-length Sgo1 with the pentameric condensin since distances between crosslinked residues on the models were all within the 25 Å reach of the BS3 crosslinker used, but shortest for Rank 5 (Figs. 3A,B and EV3D,E; Dataset EV2), indicating the best structural fit to our experimental data. The positions of residues flanking L508 F509 were also predicted with relatively high confidence, while the positions of more distant residues were not (Figs. 3C and EV3C). In all models, Sgo1 L508 and F509 fit into a hydrophobic binding pocket between HEAT repeats 3 and 4 on the concave surface under the nose of Ycg1 (Fig. 3C). In the model, Sgo1 binds to a surface on Ycg1 that is opposite the Ycg1-DNA interface. Modeling Sgo1 onto the structure of Ycg1-Brn1 complexed with DNA further suggested that Ycg1-Brn1 could bind Sgo1 and DNA concurrently (Fig. EV3F). In the predicted structure, Sgo1 L508 inserts into a binding pocket comprised of Ycg1 residues M145, I148, I151, F156, and F185 (Fig. 3D). Although Sgo1 CR1 is conserved only among yeasts closely related to *S. cerevisiae*, its predicted binding site on Ycg1 is highly conserved not only among yeasts (Fig. EV3G) but also among Ycg1/CAP-G proteins across phyla, including in mammals (Fig. 3E), suggesting it provides a common binding surface for multiple ligands. We conclude that Ycg1/CAP-G proteins harbor a conserved hydrophobic pocket which provides a binding site for Sgo1 and potentially other ligands.

## Mutation of the predicted Sgo1-Ycg1 interface disrupts their interaction in vivo

To determine whether the Sgo1-Ycg1 binding site we identified is functionally important in vivo, we generated specific point mutations in the endogenous copies of *SGO1* and *YCG1* (Fig. 4A).

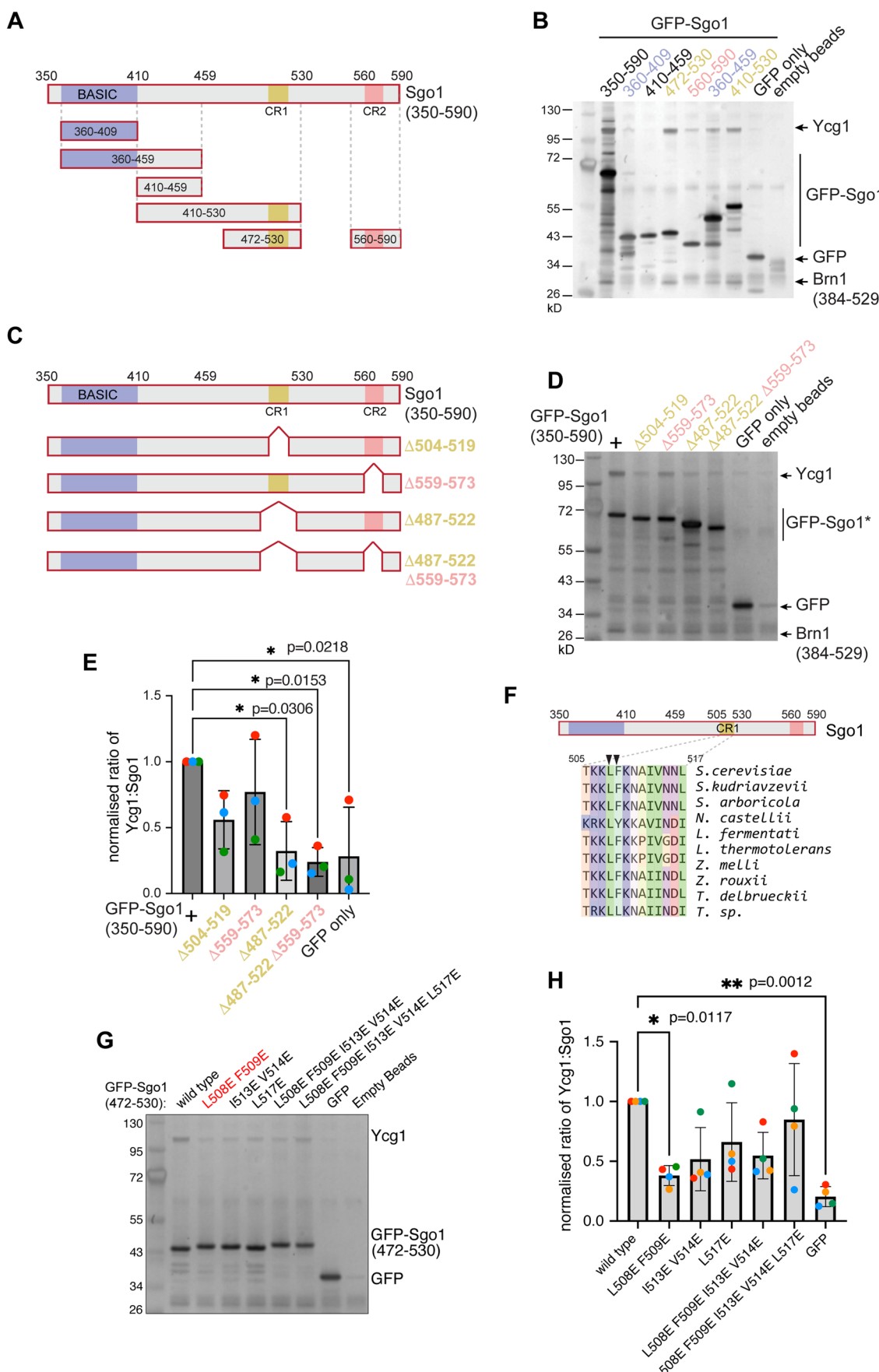

**Figure 2. A conserved region within the C-terminal region of Sgo1 is required for Ycg1 binding.**

(A, B) Recombinant GFP-tagged C-Sgo1 variants co-immunoprecipitated with Ycg1 (6–932, Δ499–555)-Brn1 (384–529) using anti-GFP-coupled beads. Scheme showing the recombinant Sgo1 C-terminal variants generated (A). The basic motif involved in Sgo1 chromosome localization is highlighted in blue. Two conserved regions found by alignment of yeast sequences are indicated in yellow (CR1) and pink (CR2). Immunoprecipitates were analyzed by silver-stained SDS-PAGE (B). Performed once as shown. (C–E) Recombinant GFP-tagged C-Sgo1 deletion mutants co-immunoprecipitated with Ycg1 (6–932, Δ499–555)-Brn1 (384–529) using anti-GFP-coupled beads. Schematic diagram of the Sgo1 recombinant C-terminal deletion mutants used in the co-immunoprecipitation assay (C). Analysis of immunoprecipitates by silver-stained SDS-PAGE (D). The ratios of Ycg1/Sgo1 gel bands intensity were calculated and the mean level from three experimental repeats with error bars representing standard deviation are shown after normalization to wild type (E). *$p < 0.0332$. In one-way ordinary ANOVA with Dunnett's correction, only comparisons significantly different from wild type are indicated. (F) Alignment of sequences from related yeast species showing the short conserved region 1 (CR1) on Sgo1. Arrowheads highlight L508 F509, mutation of these two residues showed the strongest effect in (G, H). G, H Recombinant Sgo1 (472-530) point mutants co-immunoprecipitated with Ycg1 (6–932, Δ499–555)-Brn1 (384–529), the elutes were analyzed by silver stain (G) and the mean of Ycg1/Sgo1 gel bands intensity ratio from four experimental repeats is shown after normalization to wild type (H). Error bars represent standard deviation. *$p < 0.0332$; **$p < 0.0021$, one-way ordinary ANOVA with Dunnett's correction, only comparisons significantly different from wild type are indicated. Source data are available online for this figure.

We mutated four residues M145, I148, I151, and F156 that form the hydrophobic Ycg1 pocket to alanine (*ycg1-4A*) and Sgo1 residues L508 and F509 to either alanine or glutamic acid (*sgo1-2A* and *sgo1-2E*, respectively). To understand the impact of these mutations on the Sgo1-Ycg1 interaction in vivo, we immunoprecipitated Sgo1 from metaphase-arrested cells and identified interacting proteins by mass spectrometry (Fig. 4B–F; Appendix Fig. S2A–C). In Sgo1 immunoprecipitates from wild-type cells, all five condensin subunits were abundant, and we additionally identified components of the kinetochore, cohesin and PP2A, as expected (Fig. 4B). However, comparison of Sgo1 immunoprecipitates from wild type with those from *sgo1-2A, sgo1-2E*, or *ycg1-4A* cells revealed specific depletion of condensin subunits (Fig. 4B–F), confirming the importance of the interface we identified in Sgo1-Ycg1 interaction. Importantly, similar amounts of Sgo1 were immunoprecipitated in all conditions (Appendix Fig. S2A). Furthermore, other Sgo1 interactors, including PP2A and cohesin subunits, were not significantly depleted in *sgo1-2A*, *sgo1-2E*, or *ycg1-4A* cells compared to wild type (Fig. 4B–F; Appendix Fig. S2B,C). Therefore, *sgo1-2A*, *sgo1-2E*, and *ycg1-4A* mutations specifically abrogate the Sgo1-Ycg1 interaction.

### Sgo1 CR1 is the pericentromeric receptor for condensin

During metaphase, prior to the establishment of tension-generating kinetochore-microtubule interactions, condensin associates with pericentromeres in a similar pattern to cohesin. However, unlike cohesin, condensin does not form obvious peaks on chromosome arms, and its peaks at pericentromere borders are more modest (Verzijlbergen et al, 2014; Leonard et al, 2015) (Fig. 5A). Sgo1 is also specifically localized to pericentromeres that are not under tension (Nerusheva et al, 2014; Paldi et al, 2020), in a similar pattern to condensin (Fig. 5A) and indeed the pericentromeric localization of condensin depends on Sgo1 (Yahya et al, 2020; Verzijlbergen et al, 2014). We used Brn1-6HA calibrated ChIP-Seq to assess whether direct Sgo1-Ycg1 binding is responsible for condensin recruitment to pericentromeres in metaphase-arrested cells lacking tension (using the microtubule-depolymerizing drug nocodazole to prevent the establishment of kinetochore-microtubule attachments). Western blotting confirmed that cellular Brn1 levels were unaffected by abrogation of the Sgo1-Ycg1 interaction (Fig. 5B). In wild-type cells, as expected (Verzijlbergen et al, 2014), there was a prominent Brn1 peak at all centromeres and throughout pericentromeres, although the peak height varied between chromosomes at the pericentromeric borders (Fig. 5C–F). In contrast, in *sgo1*-2A, *sgo1*-2E, and *ycg1-4A* mutants, condensin was lost throughout pericentromeres (Fig. 5C,D). Loss of Brn1 was particularly apparent at pericentromere borders (Fig. 5F), while at core centromeres, a small peak remained in all mutants (Fig. 5E). Mutations disrupting the Sgo1-Ycg1 interface have a similar effect on Brn1 localization as the complete absence of Sgo1 (Fig. 5C–F), indicating that disruption of this interface is sufficient to prevent Sgo1-dependent recruitment of condensin to pericentromeres and borders. Interestingly, residual Brn1 at centromeres in the mutants formed only a single peak, in contrast to the characteristic bimodal peak of wild-type cells (Fig. 5E). This indicates the existence of a condensin recruiter in the kinetochore, that functions through a binding surface distinct from the Ycg1 hydrophobic pocket we found to bind Sgo1. We confirmed that *sgo1-2A* and *ycg1-4A* mutations do not greatly perturb the cellular levels or localization of Sgo1 itself, while Sgo1-2E showed a general reduction in its association with chromatin for reasons that are currently unclear (Fig. EV4A–D). Therefore, disruption of the Sgo1-Ycg1 interface precludes the pericentromeric localization of condensin without affecting the pericentromeric localization of Sgo1 itself. We conclude that condensin recruitment to pericentromeres is mediated through a direct interaction with Sgo1.

### Pericentromeric condensin supports the formation of a V-shaped structure under tension

To understand how pericentromere-targeted condensin might impact chromosome organization, we performed Hi-C of wild type, *sgo1-2A*, *ycg1-4A*, and *sgo1Δ* cells arrested in metaphase either in the presence (tension) or absence (no tension) of microtubules. In the absence of tension, wild-type pericentromeres are folded into loops on either side of the centromere, while tension converts pericentromeres into a V-shape with cohesin at pericentromere borders acting as boundaries to further separation (Paldi et al, 2020) (Fig. 6A). The pericentromeric loops in the absence of tension can be visualized on centromere pile-up contact maps as a stripe which diminishes in intensity as it extends from the centromere (Fig. 6A, lower left, arrowheads). The presence of tension leads to a reduction in contacts between the centromere-proximal chromatin and the adjacent chromosome arm (Fig. 6A, upper right, dashed ovals), visualized as blue stripes on the ratio map comparing tension to no tension (Fig. 6B), together with an increase in the border strength (Appendix Fig. S3). Mutations that

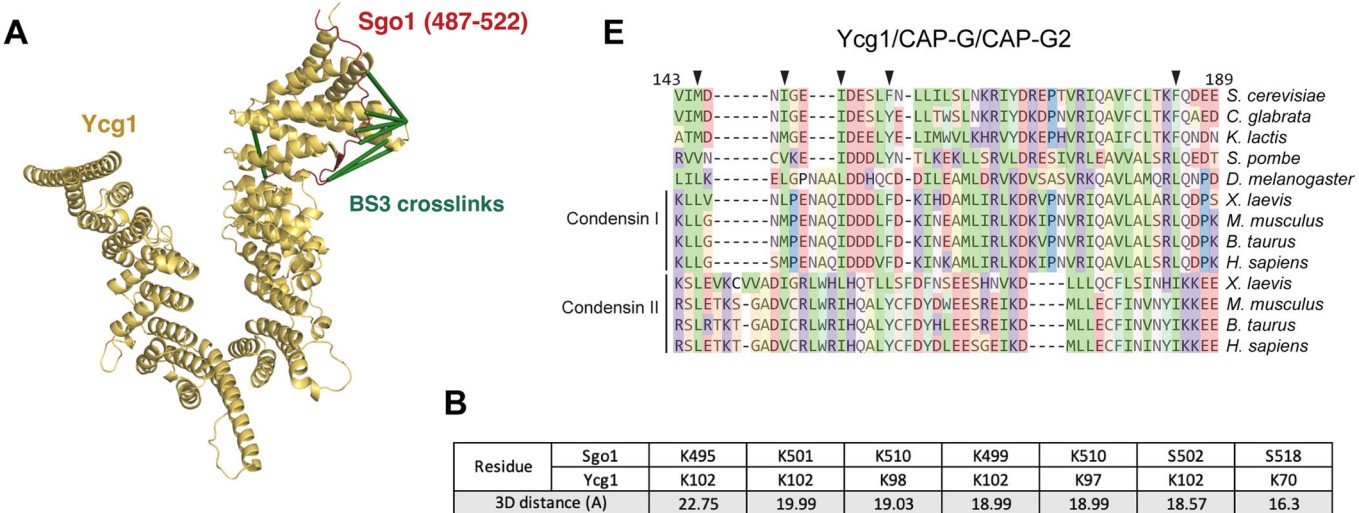

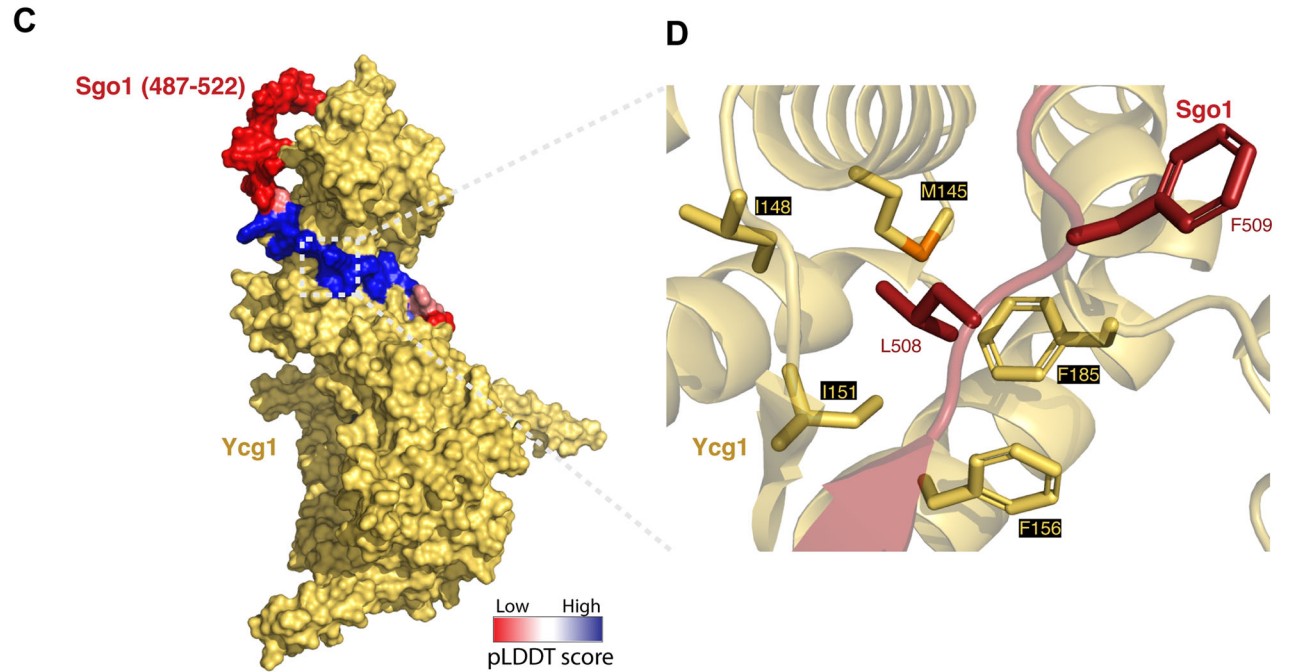

**Figure 3. Identification of a conserved binding pocket in condensin subunit Ycg1.**

(A) Mapping of crosslinks from CLMS of full-length Sgo1 and condensin holo-complex data shown in Fig. 1D onto AlphaFold2 model. (B) Distances between crosslinked residues on the AlphaFold2 model within the range of the BS3 crosslinker (25 Å). (C, D) AlphaFold2 model of Sgo1(487–522) with Ycg1 (6–932, Δ499–555)-Brn1 (384–529), Sgo1 peptide (487–522) is colored with pLDDT confidence score (C High in blue: pLDDT >70; Low in red: pLDDT <50) and with zoom (D) to show the detailed interaction surface between Sgo1 and Ycg1. (E) Conservation of the Ycg1/CAP-G binding pocket. Mutated residues are indicated with arrowheads.

abolish Sgo1 (*sgo1Δ*) or specifically abrogate the Sgo1-Ycg1 interaction (*sgo1-2A* or *ycg1-4A*) had little observable effect on pericentromere structure in the absence of tension (Fig. 6C–H, lower left). However, in the presence of tension, interactions between the pericentromere and the adjacent chromosome arm on the same side of the centromere were increased in *sgo1Δ* cells (Fig. 6E, upper right; H, upper right, blue stripes). This is likely because sister kinetochore biorientation is perturbed in *sgo1Δ* cells (Verzijlbergen et al, 2014; Peplowska et al, 2014; Indjeian et al,

2005; Indjeian and Murray, 2007), which is expected to result in failure of the pericentromere to open into its characteristic V-shape. Although much more modest, we found that *sgo1-2A* and *ycg1-4A* cells also showed slightly increased interactions between the pericentromere and adjacent arm in the tension condition (Fig. 6C,D, upper right), which was most obvious on the ratio maps (Fig. 6F,G upper right, arrowheads). There was also a modest reduction in contacts between the left and right side of the centromere within the 25 kb range (Fig. 6F,G, upper right, dashed

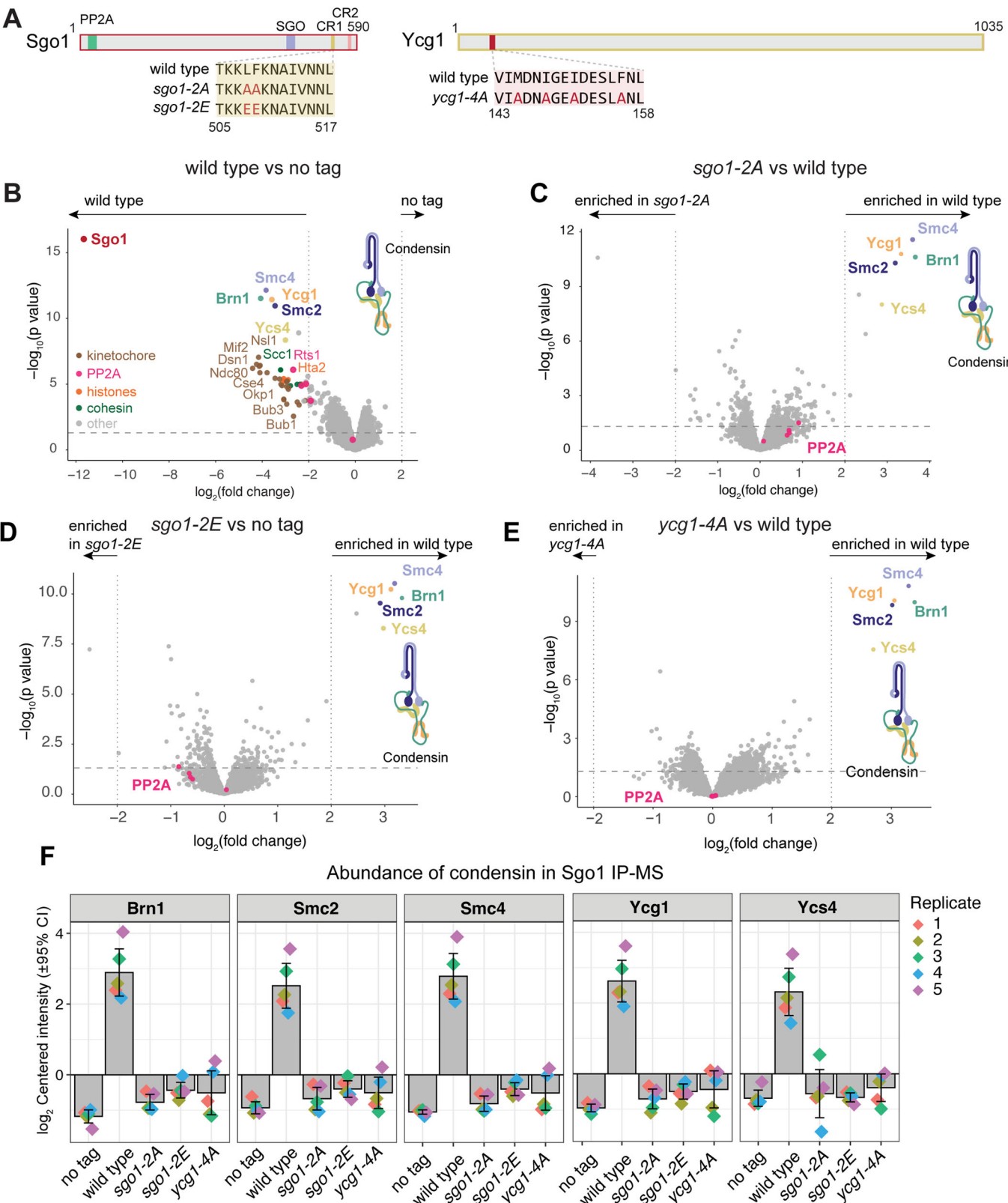

◄ **Figure 4.  Sgo1-CR1 binding to Ycg1 occurs through the conserved pocket in vivo.**

(A) Scheme showing the endogenous *SGO1* and *YCG1* point mutants generated. (B–F) Sgo1-6His-3FLAG was immunoprecipitated from mitotically-arrested cells (by benomyl treatment) and the immunoprecipitates were analysed by mass spectrometry. Volcano plots showing the relative enrichment of proteins immunoprecipitated with wild type vs no tag (B), *sgo1-2A* vs wild type (C), *sgo1-2E* vs wild type (D), and *ycg1-4A* vs wild type (E). $-\log_{10}$ (*p* values) are plotted against $\log_2$ (fold changes). Dotted line indicates $\log_2$ (fold change) = |2|. The dashed line indicates *p* value = 0.05. Protein-wise linear models and empirical Bayes statistics were used for the differential enrichment analysis. (F) Abundance of condensin subunits in Sgo1-FLAG IP-MS. Data represents values from five biological replicates. Plots show intensity values scaled to the mean of all conditions on a $\log_2$ scale and, therefore, represent relative rather than absolute comparisons. Error bars represent 95% confidence intervals. Strains used in IP-MS: no tag (AM1176), wild type (AM23137), *sgo1-2A* (AM33043), *sgo1-2E*(AM33044), and *ycg1-4A* (AM33030).

circle). The greater effect of *sgo1Δ*, compared to *sgo1-2A* and *ycg1-4A* mutations, can be attributed to the fact that Sgo1 promotes sister kinetochore biorientation through CPC-dependent error correction that is independent of condensin recruitment (Verzijlbergen et al, 2014). As expected, the strength of pericentromere borders was not affected by the shugoshin or condensin mutations (Appendix Fig. S3), consistent with cohesin, rather than condensin being responsible for boundary formation at this site (Paldi et al, 2020). The fact that we observe, albeit modest, condensin-dependent structural changes in pericentromeres only in the presence of spindle tension is surprising since condensin is removed from pericentromeres under tension (Nerusheva et al, 2014; Leonard et al, 2015). This suggests that pericentromeric condensin exerts an undetectable (at least within the resolution of our Hi-C assay) structural change in early mitosis, the functional consequence of which becomes evident only upon the establishment of sister kinetochore biorientation.

## Pericentromeric condensin facilitates sister kinetochore biorientation in mitosis

Sister kinetochore biorientation in mitosis relies on Sgo1 (Verzijlbergen et al, 2014; Indjeian et al, 2005; Kiburz et al, 2008). However, Sgo1 recruits several effectors to pericentromeres including the CPC/Aurora B, PP2A, and condensin (Verzijlbergen et al, 2014; Peplowska et al, 2014). While condensin is important for efficient biorientation (Verzijlbergen et al, 2014), the extent to which Sgo1 promotes biorientation through condensin was unclear as a mutation that specifically abolishes this interaction was not previously available. To address this, *sgo1-2E* and *ycg1-4A* cells carrying a GFP label close to the centromere of chromosome IV (*CEN4*-GFP) and a spindle pole body label (Spc42-tdTomato) were released from a G1 arrest into a metaphase arrest (by depletion of Cdc20) in the presence of nocodazole to depolymerize microtubules. Upon nocodazole wash-out, microtubules reform and subsequent sister kinetochore biorientation will generate tension resulting in the splitting of sister chromatid *CEN4*-GFP labels (Fig. 6I). We scored the percentage of cells with two GFP foci through time after nocodazole washout as a measure of the efficiency and magnitude of sister kinetochore biorientation in the population, with the percentage of cells with two Spc42-tdTomato foci serving as a control for the numbers of metaphase cells in all conditions (Fig. 6J). This revealed that the biorientation of sister kinetochores was both delayed and incomplete in *sgo1-2E* and *ycg1-4A* cells, albeit not to the extent of *sgo1Δ* cells (Fig. 6K), consistent with the effect of these mutations on pericentromere structure (Fig. 6F–H). Therefore, Sgo1 promotes sister kinetochore biorientation, in part through the recruitment of condensin to structure pericentromeres via direct interaction with Ycg1 (Fig. 6L).

## Sgo1 initiates chromosome arm condensation upon sister kinetochore biorientation

In early budding yeast mitosis, bulk chromosome condensation requires cohesin rather than condensin, which localizes to chromosome arms only upon anaphase onset, in a phosphorylation-dependent manner (Schalbetter et al, 2017; Verzijlbergen et al, 2014; Leonard et al, 2015; Renshaw et al, 2010; St-Pierre et al, 2009). Interestingly, we found that chromosome arm condensation is initiated already in metaphase, upon the establishment of sister kinetochore biorientation. Our Hi-C analysis of wild-type cells detected an increase in mid-range (10–100 kb) contacts along chromosome arms in metaphase in the presence of tension, compared to the absence of tension (Fig. 7A, dashed ovals; Appendix Fig. S4A). Since Sgo1 has also been implicated in licensing the condensation of chromosome arms in anaphase (Kruitwagen et al, 2018), we asked whether condensation in metaphase required Sgo1, its CR1 motif or the CR1-binding Ycg1 pocket (Fig. 7B–D; Appendix Fig. S4). This revealed that Sgo1 is also required for arm condensation in metaphase under tension since mid-range contacts of *sgo1Δ* cells were reduced compared to wild type (Fig. 7D, upper right, dashed ovals). Sgo1 also had a modest effect on chromosome arm condensation in the absence of tension (Fig. 7D, bottom left, solid line ovals). Therefore, Sgo1 licenses chromosome arm condensation already in metaphase, concurrent with its removal from pericentromeres. The tension-dependent condensation of chromosome arms also depended on the conserved Ycg1 patch, since it failed to occur in the *ycg1*-4A mutant (Fig. 7C, upper right, dashed ovals). Surprisingly, however, *sgo1-2A* cells were proficient in chromosome arm condensation under tension (Fig. 7B). Plotting contact probability curves confirmed these conclusions (Fig. 7E–H). Wild-type and *sgo1-2A* cells show increased contact frequency in the 10–100 kb range in the presence, compared to the absence, of tension (Fig. 7E,F, arrowheads). In contrast, contact frequencies in this range are not increased in response to tension in *ycg1-4A* and *sgo1Δ* cells (Fig. 7G,H). Therefore Sgo1, but not its direct interaction with condensin, is required for chromosome arm condensation in metaphase. Furthermore, the dependence of chromosome arm condensation on the Ycg1 patch suggests the existence of a distinct receptor capable of activating condensin in metaphase.

## A common condensin recruitment mechanism: Ycg1 conserved patch and Lrs4 CR1 motif recruit condensin to the rDNA

We considered the possibility that other ligands associate with the conserved Ycg1/CAP-G pocket in a similar way to Sgo1. Consistently, although *ycg1-5A* (M145A, I148A, I151A, F156A,

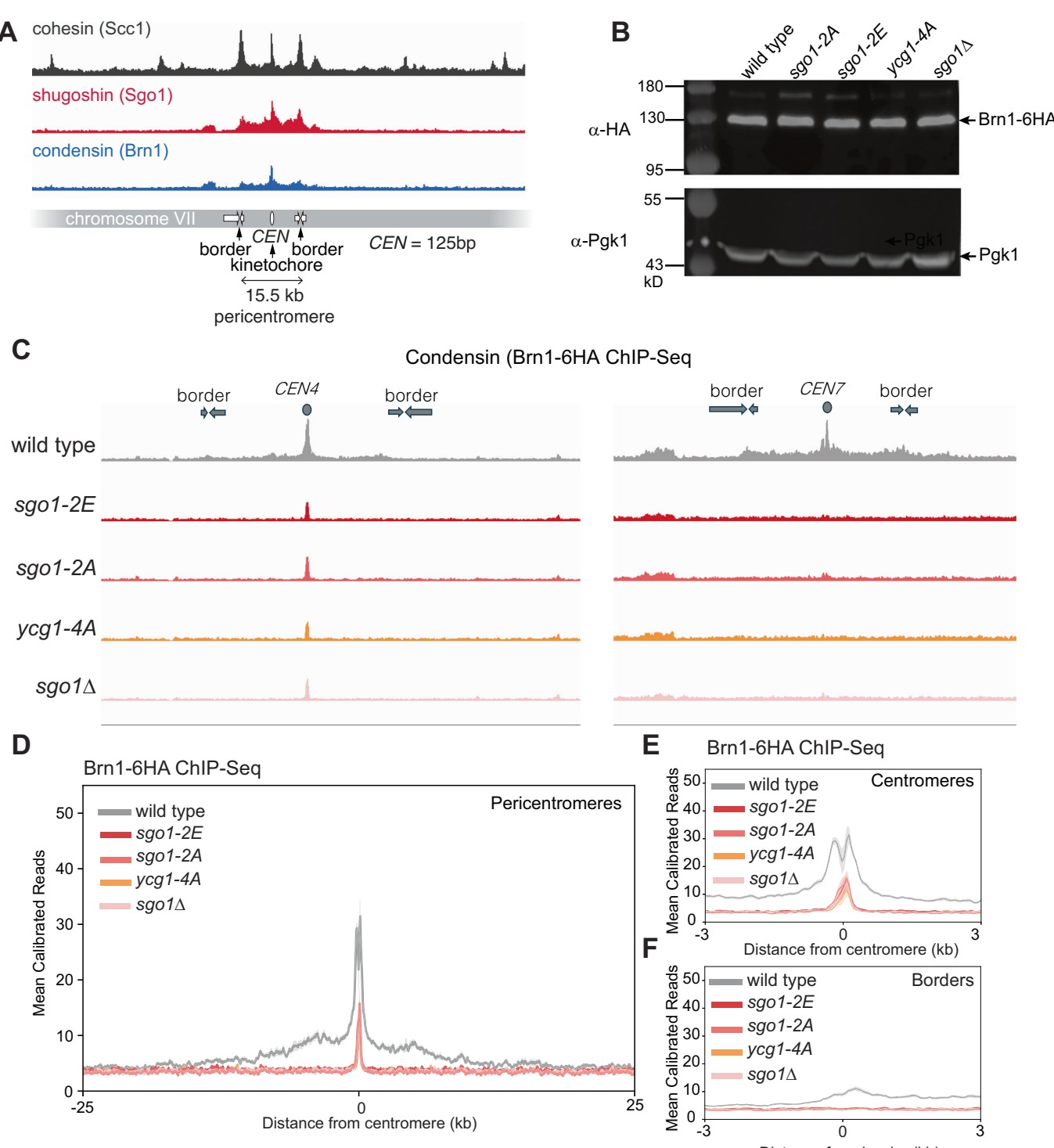

**Figure 5. Condensin binding pocket engages with Sgo1-CR1 for enrichment at pericentromeres.**

(A) Calibrated cohesin (Scc1-6HA), shugoshin (Sgo1-6His-3FLAG), and condensin (Brn1-6HA) ChIP-Seq enrichment in metaphase-arrested cells in the presence of nocodazole along chromosome VII. (B) Anti-HA immunoblot showing Brn1 is expressed in a similar level in all strains. Anti-Pgk1 immunoblot is shown as a loading control. (C–F) Calibrated condensin (Brn1-6HA) ChIP-Seq of cells arrested in metaphase by treatment with nocodazole. Condensin enrichment along representative sections of chromosome IV and chromosome VII (C), including pericentromeres are shown. Pile-ups show the mean ChIP-Seq reads (solid line) with standard error (shading) at 16 pericentromeres (D) or zoomed-in centromeres (E) and 32 borders (F). Strains used in calibrated condensin (Brn1-6HA) ChIP-Seq: *S. cerevisiae*: wild type (AM32740), *sgo1-2E* (AM33140), *sgo1-2A*(AM33145), *ycg1-4A* (AM33265), *sgo1Δ* (AM8834). *S. pombe* for calibration: AMsp635. Source data are available online for this figure.

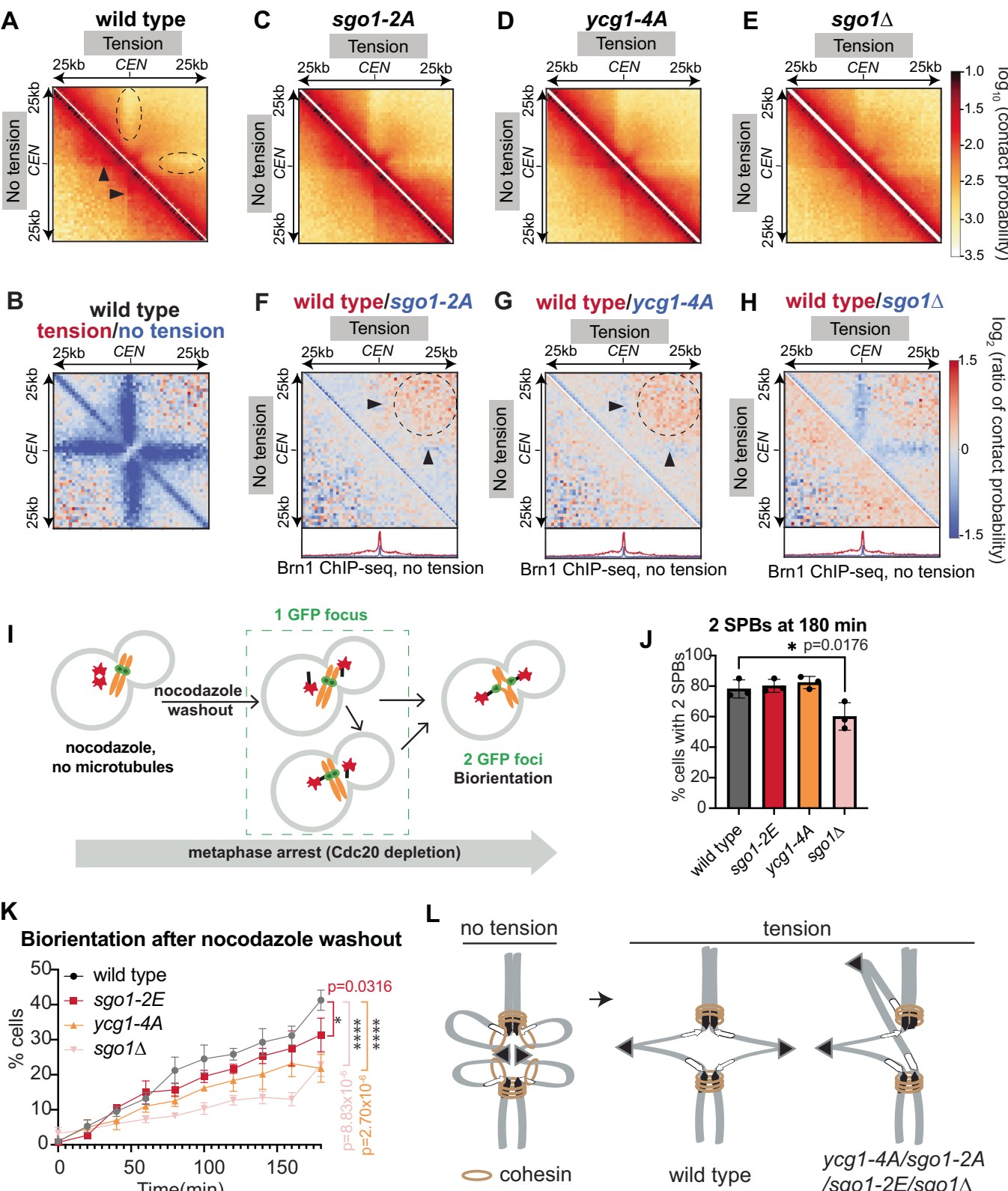

◄  **Figure 6.  Sgo1 biases sister kinetochores to biorient by condensin recruitment.**

(A–H) Hi-C of metaphase-arrested cells. In (A) and (C–H), the lower left shows a contact map of cells in the absence of spindle tension, upper right shows with tension. A, C-E share the same scale bar (beside E), B, F-H share the same scale bar (beside H). (A) Pile-ups (bin size 1 kb) of Hi-C *cis* contacts 25 kb surrounding all 16 centromeres for wild type. Scale indicates $-\log_{10}$ (contact probability) with dark orange corresponding to the highest probability (0.1) and pale yellow corresponding to the lowest contact probability (0.0003). Arrowheads indicate the stripes of pericentromeric loops on either side of the centromere in no tension cells; dashed ovals show a reduction of interactions between the pericentromere and the adjacent arm on the same side of the centromere in cells with tension. (B) Hi-C $\log_2$ ratio maps between tension (red) and no tension (blue) metaphase-arrested wild-type cells. The ratio of contact probability is shown on a $\log_2$ scale, with red indicating more frequent contacts in cells with tension and blue indicating more frequent contacts in no-tension cells. (C–E) Centromere pileups for *sgo1-2A* (C), *ycg1-4A* (D), and *sgo1Δ* (E) as in (A). (F–H) Hi-C $\log_2$ ratio maps between wild type (red) and mutants (blue) without (lower left) or with tension (upper right). Red means more frequent contacts in wild type and blue indicates more contacts in the mutant. The blue stripes (arrowheads) indicate stronger pericentromere-adjacent arm contacts in the mutant; dashed circles highlight contacts between the left and right side of the centromere within the 25 kb range in cells with tension. The corresponding condensin (Brn1-6HA) calibrated ChIP-Seq pileups are shown on the bottom (wild type in red, mutants in blue). Strains used in Hi-C: wild type (AM33174), *sgo1-2A*(AM33142), *ycg1-4A* (AM33171), and *sgo1Δ* (AM33592). (I) Schematic diagram showing the biorientation assay. Wild type, *sgo1-2E*, *ycg1-4A*, and *sgo1Δ* cells carrying *CEN4-GFP* (green) and SPB (*SPC42-tdTomato*, red) markers were released from a G1 arrest into nocodazole, and arrested in metaphase by Cdc20 depletion. After 3 h, nocodazole was washed out ($t = 0$) and *CEN4*-GFP separation was scored every 20 min for 3 h. (J) Bar chart showing the percentage of cells with two SPB dots at the last timepoint (180 min). Bar chart shows mean with error bars representing the standard deviation. *$p < 0.0332$, one-way ordinary ANOVA with Dunnett's correction, only comparisons significantly different from wild type are indicated, $n = 100$–200. (K) The percentage of separated *CEN4-GFP* foci across the time-course are shown. Error bars represent the standard error of the mean. $p$ values refer to analysis of the last time point, *$p < 0.0332$, ****$p < 0.0001$, two-way ANOVA, with Tukey correction, only comparisons significantly different from wild type are indicated. Typically, 200 cells (at least 100 cells) were scored for each timepoint. Strains used in the biorientation assay: wild type (AM33464), *sgo1-2E*(AM33466), *ycg1-4A* (AM33167), *sgo1Δ* (AM6117). (L) Model showing the effects of impaired sister kinetochore biorientation on pericentromere structure in the *ycg1-4A/sgo1-2A/sgo1-2E/sgo1Δ* cells. Source data are available online for this figure.

and F185A) mutants are viable, indicating that the essential condensin function is intact, they form smaller colonies than either wild type or sgo1-2E mutants (Fig. EV5A), suggesting that binding of other ligands to the Ycg1 hydrophobic pocket influences growth. One candidate Ycg1 ligand is Lrs4, which is required for condensin positioning at the rDNA and mating locus (Dinda et al, 2023; Johzuka and Horiuchi, 2009) and which we found to harbor a conserved CR1-like motif (Fig. 8A). Indeed, alphafold modeling of an Lrs4 peptide placed its CR1-like motif in the conserved Ycg1 patch, with residue L322 making nearly identical contacts to Sgo1 L508 (Fig. 8B,C). Mutation of the Lrs4 CR1 motif, either by substitution of L322 to alanine or glutamic acid (lrs4-L322A or lrs4-L322E), or by mutating the preceding four basic residues to alanine (lrs4-4A) resulted in a loss of condensin enrichment in the rDNA (Fig. EV5B–E). Similarly, Brn1 localization to the rDNA repeats was abrogated in ycg1-4A, but not sgo1-2A, sgo1-2E or sgo1Δ cells (Fig. 8D). Accordingly, condensation of the rDNA under tension was specifically reduced in ycg1-4A mutants (Fig. 8F, arrowhead; Appendix Fig. S5C), not in sgo1-2A or sgo1Δ cells (Fig. 8E,G; Appendix Fig. S5B,D). Therefore, condensation of the rDNA requires the targeting of condensin through the Ycg1 conserved patch and Lrs4-CR1 motif, similar to Sgo1-dependent recruitment of condensin to pericentromeres. Furthermore, analysis of the lrs4-4A mutations provides evidence that basic residues in the CR1 motif contribute to the binding of the Ycg1 conserved pocket.

Interestingly, the residues within the Sgo1 CR1 motif involved in binding to Ycg1 are reminiscent of the sequence in the tail of KIF4A previously found to be required for association with condensin (Wang et al, 2020; Takahashi et al, 2016) (Fig. EV5F). To identify further potential condensin ligands that could associate with the conserved Ycg1/CAP-G patch, we scanned proteins of interest for consensus CR1-like motifs derived from comparing Sgo1 and Lrs4 ([KR]-[KR]-L-[FYVIT]-[KR]-x(1,3)-[IV]). This revealed the presence of a CR1-like motif in several chromosome-associated budding yeast proteins, providing candidate condensin ligands for future studies (Fig. EV5G). Together, these observations raise the interesting possibility that multiple ligands use CR1-like motifs to dock onto the conserved Ycg1/CAP-G binding pocket.

## Discussion

We have presented a molecular explanation for how condensin activity is targeted to chromosomal subdomains to direct context-specific folding underlying their functions. We demonstrate that the HAWK subunit Ycg1 uses a conserved binding pocket to dock onto a conserved motif, CR1, which is present in a number of known and potential condensin regulators. We show that the Ycg1 conserved pocket docks onto distinct chromosomal receptors to structure pericentromeres, rDNA, and chromosome arms and that the interplay between them leads to an ordered structuring of chromosomes as cells transition through mitosis (Fig. 8H).

Our findings have implications for how condensin engages with ligands more generally. The binding pocket we identified on Ycg1 is broadly conserved and we discovered CR1-like motifs in the known condensin ligands yeast Sgo1 and Lrs4, and mammalian KIF4A, together with several potential ligands that we identified by homology search of the yeast proteome. The functional relevance of KIF4A as a CAP-G binding ligand was recently demonstrated (Cutts et al, 2024). While future work is required to test whether these candidates are true condensin ligands and to determine their functions, it is likely that additional chromosomal receptors use the same mode of condensin binding to define the architecture of chromatin subdomains.

### Specific targeting of SMC proteins to pericentromeres

The structuring of pericentromeres by cohesin and condensin allows them to perform specialized functions in chromosome segregation (McAinsh and Marston, 2022). Our explanation of the specific targeting of condensin by Sgo1 adds to our understanding of how pericentromere architecture is defined. Initially, cohesin is recruited to kinetochores by Ctf19, resulting in the formation of a loop on either side of the centromere (Hinshaw et al, 2017; Paldi et al, 2020). Sgo1, in turn, requires cohesin, as well as Bub1-dependent phosphorylation of H2A-S121 (Thr120 in humans) for its pericentromeric localization (Verzijlbergen et al, 2014; Kawashima et al, 2010; Nerusheva et al, 2014; Liang et al, 2020; Liu et al, 2013b, 2013a). Finally, Sgo1 engages condensin to promote proper

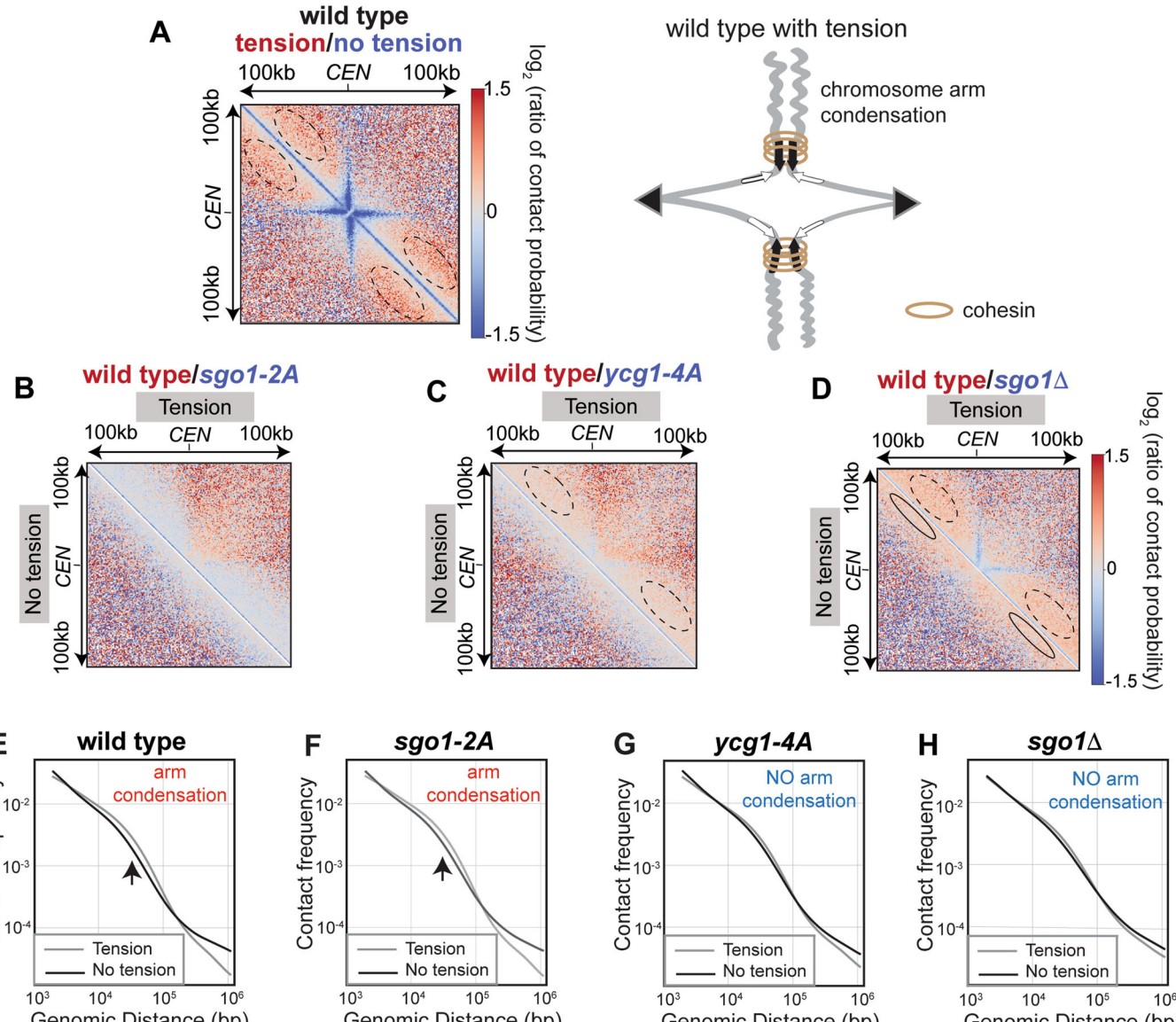

**Figure 7.  Sgo1 initiates chromosome arm condensation in metaphase.**

(**A**) Hi-C log₂ ratio maps between tension (red) and no tension (blue) metaphase-arrested wild-type cells. The ratio map is made from the pileups (bin size 1 kb) of *cis* contacts 100 kb surrounding all 16 centromeres. The ratio of contact probability is shown on a log₂ scale, with red indicating more frequent contacts in cells with tension and blue indicating more contacts without tension. Dashed ovals indicate an increase in mid-range (10–100 kb) arm contacts in wild-type cells with tension, compared to without tension. The model on the right shows chromosome arm condensation in metaphase in wild-type cells with tension. (**B–D**) Hi-C log₂ ratio maps between wild type (red) and mutants (blue) along chromosome arms. The lower left triangle of the heatmap shows metaphase-arrested cells without spindle tension and upper right shows cells with tension. They share the same scale bar (besides **D**). Red indicates more frequent contacts in the wild type and blue means more frequent contacts in the mutants. The reduction of mid-range contacts in *ycg1-4A* (**C**) and *sgo1Δ* (**D**) cells with tension is highlighted with dashed ovals. Solid line ovals indicate a reduction in contacts in *sgo1Δ* without tension (**D**). (**E–H**) *P(s)* curve for metaphase-arrested cells without spindle tension (black) and with tension (gray) averaged over all chromosomes. Arrowheads show that the mid-range contacts increase in the presence of tension in wildtype and *sgo1-2A* cells (chromosome arm condensation in metaphase in the presence of tension), but not in *ycg1-4A* and *sgo1Δ* cells (No arm condensation). Strains used in Hi-C: wild type (AM33174), *sgo1-2A* (AM33142), *ycg1-4A* (AM33171), and *sgo1Δ* (AM33592). Source data are available online for this figure.

sister kinetochore biorientation. Sgo1 also ensures biorientation by localizing the chromosome passenger complex (CPC) containing Aurora B at kinetochores, which destabilizes tension-less kinetochore-microtubule attachments to provide a further opportunity for correct attachments to be made (Verzijlbergen et al, 2014; Peplowska et al, 2014; Broad et al, 2020; Hadders et al, 2020) Upon the establishment of biorientation, loop-extruding cohesin,

Sgo1, condensin, and CPC are all released from pericentromeres, while cohesive cohesin is retained at pericentromere borders to resist pulling forces (Paldi et al, 2020; Nerusheva et al, 2014; Leonard et al, 2015). Interestingly, we found evidence that inter-sister kinetochore tension in metaphase also initiates the condensation of chromosome arms. This requires Sgo1, but not its binding to Ycg1. The CPC is important for chromosome

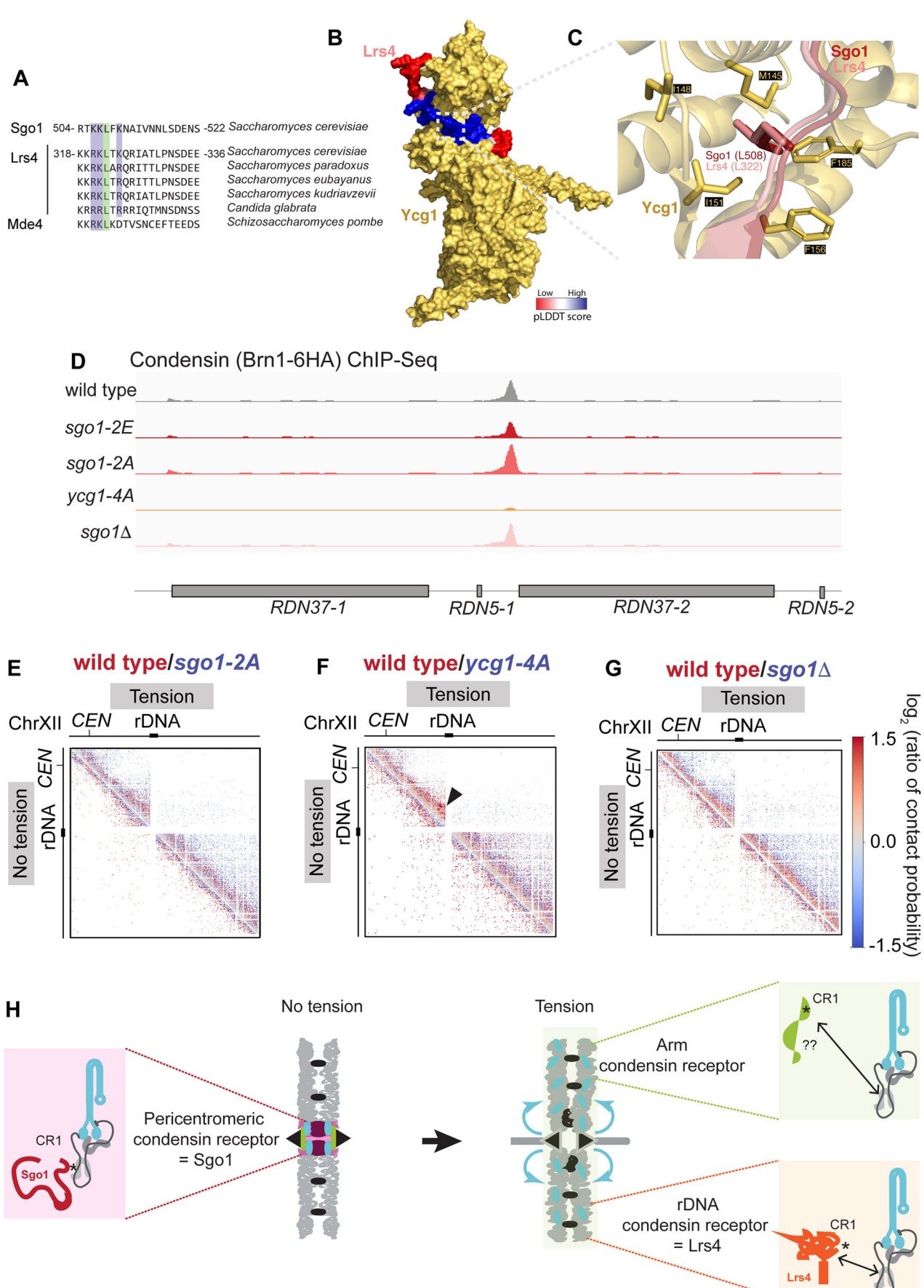

**A**

Sgo1    504- RTK**K**L**F**KNAIVNNLSDENS -522  *Saccharomyces cerevisiae*

Lrs4    318- KK**R**K**L**T**K**QRIATLPNSDEE -336  *Saccharomyces cerevisiae*
         KK**R**K**L**A**R**QRITTLPNSDEE       *Saccharomyces paradoxus*
         KK**R**K**L**T**R**QRITTLPNSDEE       *Saccharomyces eubayanus*
         KK**R**K**L**T**R**QRIATLPNSDEE       *Saccharomyces kudriavzevii*
         KK**R**R**L**T**R**RIQTMNSDNSS        *Candida glabrata*

Mde4    KK**R**K**L**KDTVSNCEFTEEDS            *Schizosaccharomyces pombe*

**B** Lrs4 / Ycg1

Low — High
pLDDT score

**C** Sgo1 / Lrs4 / Ycg1
M145, I148, I151, F156, F185
Sgo1 (L508) / Lrs4 (L322)

**D** Condensin (Brn1-6HA) ChIP-Seq

wild type
*sgo1-2E*
*sgo1-2A*
*ycg1-4A*
*sgo1Δ*

*RDN37-1*   *RDN5-1*   *RDN37-2*   *RDN5-2*

**E** wild type/*sgo1-2A*

**F** wild type/*ycg1-4A*

**G** wild type/*sgo1Δ*

ChrXII  *CEN*  rDNA   Tension

No tension  rDNA  *CEN*

$\log_2$ (ratio of contact probability)
1.5
0.0
-1.5

**H**

No tension          Tension

Pericentromeric condensin receptor = Sgo1

Arm condensin receptor
CR1  ??

rDNA condensin receptor = Lrs4
Lrs4  CR1

**Figure 8.   General principles of condensin targeting.**

(A) A CR1-like motif is found in the monopolin protein Lrs4/Mde4. (B) AlphaFold2 model of Ycg1 (yellow) with Lrs4 colored with pLDDT confidence score (High in blue: pLDDT >70; Low in red: pLDDT <50). (C) The detailed interaction surface, Lrs4 (pink) aligns with Sgo1 (red), sticks highlight the hydrophobic binding pocket. (D) Calibrated condensin (Brn1-6HA) ChIP-Seq of nocodazole arrested cells at the rDNA region. Strains used: *S. cerevisiae*: wild type (AM32740), *sgo1-2E* (AM33140), *sgo1-2A*(AM33145), *ycg1-4A* (AM33265), *sgo1Δ* (AM8834). *S. pombe* for calibration: AMsp635. (E–G) Hi-C log$_2$ ratio maps between wild type (red) and mutants (blue) along Chromosome XII. Cells are metaphase-arrested in the absence (lower left) and in the presence (upper right) of tension. They share the same scale bar (beside G). Red means more frequent contacts in the wildtype and blue means more frequent contacts in the mutant. The reduction of rDNA condensation in *ycg1-4A* mutants under tension is highlighted with an arrowhead (upper right of F). Strains used in Hi-C: wild type (AM33174), *sgo1-2A*(AM33142), *ycg1-4A* (AM33171), and *sgo1Δ* (AM33592). (H) Model showing general principles of condensin targeting in mitosis. In the absence of spindle tension, Sgo1 recruits condensin to pericentromeres. Once tension is generated, Lrs4 brings condensin to the rDNA, and a third unknown receptor recruits condensin to chromosome arms. They are all regulated through direct binding between receptor CR1 motif and a conserved binding pocket of HAWK subunit Ycg1.

condensation (Lavoie et al, 2004; Giet and Glover, 2001; Petersen and Hagan, 2003) and understanding the interplay between Sgo1, CPC and condensin in biorientation and chromosome condensation is an important priority.

How might pericentromeric condensin promote sister kinetochore biorientation? Chromosome capture experiments in budding yeast and vertebrates have observed structural changes at centromeres in condensin-deficient cells (Schalbetter et al, 2017; Sacristan et al, 2022). Consistently, several studies have suggested that condensin provides a rigidity that allows pericentromeres to resist pulling forces and establish tension, which in turn is required to prevent kinetochore-microtubule destabilization by the error correction machinery (Ribeiro et al, 2009; Samoshkin et al, 2009; Nakazawa et al, 2008; Oliveira et al, 2005). An alternative, but not mutually exclusive possibility, is that condensin provides a geometry to pericentromeres that positions sister kinetochores in a back-to-back orientation, providing an intrinsic bias for sister kinetochores to attach to microtubules from opposite poles. In budding yeast, where a single microtubule contacts each kinetochore, such a bias has been observed to exist and requires both Sgo1 and condensin, in support of this idea (Verzijlbergen et al, 2014; Kiburz et al, 2008; Indjeian and Murray, 2007). One possibility is that dynamic loop extrusion by pericentromeric condensin leads to the formation of nested loops of different sizes within the cohesin-generated pericentromeric loops. This could confer both structural rigidity and the appropriate geometry to pericentromeres. Understanding the interplay between cohesin and condensin in defining pericentromere structure before and after tension establishment and how it facilitates chromosome segregation is an important question for the future.

Condensin is also enriched at vertebrate pericentromeres where it plays an important role in providing structural rigidity (Ribeiro et al, 2009; Sacristan et al, 2022). Whether shugoshins also play a role in condensin recruitment to pericentromeres in other organisms is not known, however we found no obvious CR1 motif in vertebrate shugoshins. Potentially other CR1-containing proteins concentrate condensins in pericentromeres in vertebrates. Alternatively, vertebrate shugoshins may harbor a cryptic CR1 motif that is not easily recognizable by sequence homology.

## Commonalities in regulation of SMC proteins by ligand-binding HAWKs

Our discovery of a conserved ligand-binding pocket on the condensin HAWK Ycg1, is reminiscent of the Conserved Essential Surface (CES) on cohesin. The CES is formed of a composite surface between the HAWK, SA2, and the kleisin RAD21 and is a binding site for multiple regulators (Li et al, 2020; García-Nieto et al, 2023). These include the loop anchor protein CTCF, the kinetochore protein CENP-U and SGO1, all of which compete with the cohesin release factor WAPL that also binds to this site (Li et al, 2020; García-Nieto et al, 2023; Yan et al, 2024). Interestingly, the CES occupies a similar position on SA2 to the conserved pocket on Ycg1 we identified here, suggesting that general principles may regulate SMC protein complexes.

## Methods

### Reagents and tools table

| Reagent/resource | Reference or source | Identifier or catalog number |
|---|---|---|
| **Experimental models** | | |
| See Table EV1 | | |
| **Recombinant DNA** | | |
| See Table EV2 | | |
| **Antibodies** | | |
| Mouse monoclonal anti-GFP | Roche | Cat#11814460001 |
| Mouse monoclonal anti-V5 | Biorad | Cat#MCA1360 |
| Mouse monoclonal anti-HA HA11 | Biolegend | Cat#MMS-101R |
| Mouse monoclonal anti-HA 12CA5 | Roche | Cat#11666606001 |
| Mouse monoclonal anti-FLAG | Sigma | Cat#F1804 |
| **Chemicals, enzymes and other reagents** | | |
| BaseMuncher Endonuclease | Abcam | Cat#ab270049 |
| Benzonase | Merck Millipore (Novagen) | Cat#71206-3 |
| Protein G Dynabeads | Thermo Fisher Scientific | Cat#10009D |
| Chymostatin | Melford | Cat#C1104 |
| Leupeptin (Hemisulphate) | Melford | Cat#L1001 |
| E-64 | Melford | Cat#E1101 |

| Reagent/resource | Reference or source | Identifier or catalog number |
|---|---|---|
| Pepstatin A | Melford | Cat#P2203 |
| Antipain, dihydrochloride | Melford | Cat#A0105 |
| Aprotinin | Melford | Cat#A2301 |
| AEBSF hydrochloride 98% | ACROS Organics | Cat#32811010 |
| N-Ethylmaleimidine 99+% | ACROS Organics | Cat#156100050 |
| Complete EDTA-free tablets | Roche | Cat#11873580001 |
| Microcystin-L | LKT Laboratories | Cat#M3406 |
| NP-40 | Thermo Fisher Scientific | Cat#13444269 |
| NuPAGE 4–12% Bis-Tris gel | Invitrogen | Cat#NP0323BOX |
| NuPAGE 3–8% Tris-Acetate gel | Invitrogen | Cat#EA0378BOX |
| DMP (Dimethyl Pimelimidate) | Sigma | Cat#D8388 |
| BS3 (bis(sulfosuccinimidyl) suberate) | Thermo Fisher Scientific | Cat#A39266 |
| EDC (1-ethyl-3-(3-dimethylaminopropyl)) | Thermo Fisher Scientific | Cat#22980 |
| NHS (N-hydroxysulfosuccinimide) | Thermo Fisher Scientific | Cat#24510 |
| Trypsin | Pierce | Cat#90057 |
| Glu-C Endoproteinase | Thermo Fisher Scientific | Cat#90054 |
| Rapigest | Waters | Cat#186001861 |
| Luna Universal qPCR Master Mix | NEB | Cat#M3003X |
| Silver stain | Invitrogen | Cat#LC6070 |
| ECL SuperSignal West Pico chemiluminescence kit | Thermo Fisher Scientific | Cat#34580 |
| 2100 Bioanalyzer High Sensitivity DNA kit | Agilent | Cat#5067-4626 |
| MiniSeq High output reagent kit (150-cycles) | Illumina | Cat#FC-420-1002 |
| NEXTflex-6 DNA Barcodes | PerkinElmer | Cat#v514101 |
| Qubit HS DNA assay kit | Thermo Fisher Scientific | Cat#Q32854 |
| **Software** | | |
| DIA-NN | Demichev et al, 2020 | N/A |
| R studio | Posit Software | https://posit.co/download/rstudio-desktop |
| Prism 10 | Graphpad | https://www.graphpad.com/features |
| Adobe Illustrator | Adobe | https://www.adobe.com/uk/ |

| Reagent/resource | Reference or source | Identifier or catalog number |
|---|---|---|
| JalView | Waterhouse et al, 2009 | https://www.jalview.org |
| AlphaFold2 | Evans et al, 2022; Jumper et al, 2021; Mirdita et al, 2022 | https://colab.research.google.com/github/sokrypton/ColabFold/blob/main/AlphaFold2.ipynb |
| PyMOL | Schrödinger, LLC | https://www.pymol.org/ |
| **Other** | | |
| Illumina MiniSeq instrument | Illumina, San Diego, CA | |

## Yeast strains and plasmids

All yeast strains are W303 derivatives and are listed in Table EV1. *SGO1-FLAG*, *BRN1-HA*, and *CEN4-GFP* were described previously (Verzijlbergen et al, 2014). *SGO1* and *LRS4* mutants were introduced using standard PCR-based methods. *YCG1-4A* was made by CRISPR-Cas9 in this study.

Plasmids used and generated in this study are listed in Table EV2. *SGO1* fragments, deletion, and point mutant variants were generated using PCR and cloned using Gibson assembly. *GFP-SGO1* used in in *E. coli* expression was codon optimized, and *V5-SGO1* used in yeast expression was described previously (Su et al, 2021). Constructs for the generation of pentameric condensin complex and *YCG1* (6–932, Δ499–555) - *BRN1* (384–529) were kind gifts from Damien D'Amours (St-Pierre et al, 2009) and Christian Haering (Kschonsak et al, 2017).

### Protein expression and purification

Expression and purification of V5-Sgo1 (FL*)* (used in co-immunoprecipitation and crosslinking mass spectrometry, Figs. 1B–D; EV1A) was described previously (Su et al, 2021). Briefly, the protein was expressed in a protease-deficient yeast strain (AM8184) with 0.5 mM $CuSO_4$ for 6 h. After ball breaker grinding, cell powder was resuspended in lysis buffer (25 mM Tris-HCl, pH 7.5, 150 mM NaCl, 1 mM $MgCl_2$, 10% glycerol, 0.1% NP-40, 0.05 mM EDTA, 0.05 mM EGTA, and 1 mM DTT) plus protease inhibitors (1× CLAAPE [chymostatin, leupeptin, aprotinin, antipain, pepstatin, and E-64], 1 mM Pefabloc, 0.4 mM Na orthovanadate, 0.1 mM microcystin, 1 mM *N*-ethylmaleimide, 2 mM β-glycerophosphate, 1 mM Na pyrophosphate, 5 mM NaF and complete EDTA-free protease inhibitor (Roche)), and treated with 40 U/ml benzonase for 1.5 h. The crude lysate was diluted with 25 mM Na phosphate, pH 7.5, 500 mM NaCl, 10% glycerol, 0.1% NP-40, 10 mM imidazole, 1 mM DTT, and 0.25 mM PMSF, and after ultracentrifugation and filtering, loaded onto $Ni^{2+}$ conjugated HiTrap IMAC FF 1-ml column (GE Healthcare). After washing with 25 mM Na phosphate, pH 7.5, 500 mM NaCl, 10% glycerol, 0.1% NP-40, 25 mM imidazole, 1 mM DTT and 0.25 mM PMSF, the protein was eluted with an increasing imidazole gradient (25–500 mM) over 40 column volumes and then loaded onto a gel filtration Superose 6 10/300 column in 50 mM Tris-HCl, pH 7.5, 500 mM NaCl, 10% glycerol, 0.1 mM EDTA, 0.1 mM EGTA, 1 mM DTT, and 0.25 mM PMSF.

V5-Sgo1 (350–590) (used in co-immunoprecipitation and crosslinking mass spectrometry, Fig. EV2A–E) was expressed in a protease-deficient yeast strain (AM8184) with 2% (w/v) galactose for 8 h. The protein purification was the same as V5-Sgo1 (FL).

GFP-Sgo1 fragments (used in co-immunoprecipitation assays, Fig. EV1C) were expressed in *E. coli* BL21(DE3) with 0.1 mM IPTG, at 18 °C overnight. Cells were resuspended in lysis buffer (20 mM Na phosphate pH 8.0, 500 mM NaCl, 1 mM DTT, 10 mM imidazole, 10% glycerol with complete EDTA-free protease inhibitor). After cell disruption (25kPis) and ultracentrifugation, extracts were loaded onto Ni$^{2+}$conjugated HiTrap IMAC FF 1-ml column (GE Healthcare) and proteins were eluted with 25–500 mM imidazole over 40 column volumes. The eluted protein was then loaded onto a gel filtration Superdex200 16/600 column in 50 mM Tris-HCl pH 8.0, 500 mM NaCl, 1 mM DTT, 0.25 mM PMSF, 10% glycerol. For purification of GFP-Sgo1 C-terminal fragments, deletion and point mutant variants (used in co-immunoprecipitation assays, Fig. 2), cells were resuspended in lysis buffer (25 mM HEPES, pH 7.5, 500 mM NaCl, 20 mM imidazole, 1 mM DTT, 10% glycerol with complete EDTA-free protease inhibitor). After sonication and ultracentrifugation, lysates were incubated with Ni-NTA resin for 1 h. The protein was then eluted with 200 mM imidazole, and dialyzed to no imidazole buffer.

Pentameric condensin complex (purified from yeast, used in co-immunoprecipitation, SEC and crosslinking mass spectrometry, Figs. 1B–D and EV1A,B, 2A) and Ycg1 (6–932, Δ499–555) - Brn1 (384–529) (purified from *E. coli*, used in co-immunoprecipitation and crosslinking mass spectrometry, Figs. 2 and EV2B–E) were expressed and purified as described previously (Kschonsak et al, 2017).

### Size exclusion chromatography

For full-length Sgo1 and condensin holo-complex interaction studies (Fig. EV1A), Superose6 3.2/30 2.4 ml column was used with buffer 20 mM Tris pH 7.5, 200 mM NaCl, 0.1 mM EGTA, 1 mM DTT, and 0.05 ml fractions were collected. For Sgo1 (350–590) and Ycg1 (6–932, Δ499–555)-Brn1 (384–529) interaction studies (Fig. EV2B), Superdex200 10/300 24 ml column was used, with buffer 10 mM Tris-HCl pH 7.5, 250 mM NaCl, 5% glycerol, 0.001% Tween-20, and 0.3 ml fractions were collected.

### Co-immunoprecipitation, western blotting, silver stain

To conjugate anti-V5, anti-GFP, anti-HA or anti-FLAG to dynabeads, pre-washed (0.1 M Na phosphate, pH 7.0) protein G dynabeads (Invitrogen) were incubated with mouse anti-V5 (Biorad; MCA1360), anti-GFP (Roche; 1184460001), anti-HA (Biolegend; MMS-101R), or anti-FLAG antibody (Sigma; F1804) with gentle agitation for 25 min. After washing twice with 0.1 M Na phosphate, pH 7.0, 0.01% Tween-20, then twice with 0.2 M triethanolamine, pH 8.2, proteins were crosslinked with 20 mM DMP (Dimethyl pimelimidate, Sigma) with rotation for 30 min. After quenching with 50 mM Tris-HCl, pH7.5, beads were washed three times with PBST (0.1% Tween-20).

For co-immunoprecipitation using two recombinant proteins (Figs. 1B,C; 2 and EV2A,C), pre-washed (25 mM HEPES pH 7.5, 500 mM NaCl, 10% glycerol, 1 mM DTT, and 0.1% NP-40) antibody coupled dynabeads were incubated with two purified recombinant proteins with rotation for 2.5 h. After washing five times with the same pre-wash buffer, proteins were eluted from the beads by boiling at 65 °C for 15 mins in 1x NuPAGE LDS sample buffer.

For co-immunoprecipitation using one recombinant protein and yeast extract (Fig. EV1C), cycling *sgo1Δ* yeast cells AM827 (no tag) and AM8834 (*BRN1-6HA*) were snap frozen as small "noodles" and lysed by grinding. After resuspending with lysis buffer (25 mM HEPES, pH 7.5, 2 mM MgCl$_2$ 15% glycerol, 0.1% NP-40, 150 mM KCl, 0.1 mM EDTA, and 0.5 mM EGTA) plus protease inhibitors (1× CLAAPE [chymostatin, leupeptin, aprotinin, antipain, pepstatin, and E-64], 1 mM Pefabloc, 0.4 mM Na orthovanadate, 0.1 mM microcystin, 1 mM *N*-ethylmaleimide, 2 mM β-glycerophosphate, 1 mM Na pyrophosphate, 5 mM NaF, and complete EDTA-free protease inhibitor), the crude lysates were treated with 40 U/ml of benzonase (Novagen) or BaseMuncher (Abcam) for 1 h and centrifuged at 3600 rpm for 20 min. Pre-washed anti-GFP coupled beads were incubated with recombinant GFP-Sgo1 fragments while rotating for 1.5 h, and then washed three times with lysis buffer plus 0.25 mM PMSF. These Sgo1-beads were then incubated with the yeast lysate supernatant (*BRN1-6HA*) for another 1.5 h. Following washing five times with lysis buffer plus 0.25 mM PMSF, proteins were eluted from the beads by heating at 65 °C for 15 min in 1x NuPAGE LDS sample buffer.

For western blotting, proteins were separated in SDS-PAGE gels and transferred to nitrocellulose membranes. All antibodies were diluted in 2% milk PBST. Signals were detected by Pico-ECL (Thermo Fisher Scientific) and autoradiograms.

For silver stain, proteins were separated in NuPAGE 4–12% Bis-Tris gel (Invitrogen) and stained with SilverQuest Silver Staining Kit (Invitrogen).

### Crosslinking mass spectrometry

For BS3 crosslinking of full-length Sgo1 and condensin holo-complex (Fig. 1D; Dataset EV1), 1:1 (w:w) gel filtrated proteins were incubated 1 h on ice. After buffer exchanging to 25 mM HEPES, pH7.5, 150 mM NaCl, 5% glycerol, 1 mM DTT, the protein complex was crosslinked with BS3 (Thermo Scientific, BS3:protein ratio = 3:1, w:w) for 2 h on ice. The crosslinking was quenched by 100 mM ammonia bicarbonate and was briefly resolved using a NuPAGE 3–8% Tris-Acetate gel (Invitrogen, EA0378BOX). Bands were visualized by quick InstantBlue staining (Abcam), excised, reduced with 8 mM TCEP for 20 min at room temperature, alkylated with 5 mM iodoacetamide for 20 min at room temperature, digested at 37 °C using GluC (Thermo, GluC:protein ratio = 1:10, w:w in 10 mM ammonia bicarbonate) overnight, and then with 13 ng/μl trypsin (Pierce) at 37 °C for another 8 h.

For BS3 crosslinking of Sgo1 (350–590) and Ycg1 (6–932, Δ499–555)-Brn1 (384–529) (Fig. EV2D; Dataset EV1), purified proteins (Sgo1: Ycg1-Brn1 molar ratio = 1.5:1) were incubated 1 h on ice in 25 mM HEPES pH7.5, 500 mM NaCl, and 10% glycerol. After crosslinking with BS3 (Thermo Scientific, BS3:protein ratio = 2:1, w:w) for 2 h on ice, proteins were resolved using a NuPAGE 4–12% Bis-Tris gel (Invitrogen). Similar steps were performed as above, except proteins were digested only with 13 ng/μl trypsin (Pierce) at 37 °C overnight.

For EDC crosslinking of Sgo1 (350–590) and Ycg1 (6–932, Δ499–555)-Brn1 (384–529) (Dataset EV1), 10 μg of protein complex at a Sgo1: Ycg1-Brn1 molar ratio of 3:1 was crosslinked with 10 μg EDC (Thermo Fisher Scientific) and 22 μg NHS (Thermo Fisher Scientific) in 25 mM HEPES pH 6.8, 500 mM NaCl, 10% glycerol for 1.5 h at room temperature.

LC-MS/MS analysis was performed on an Orbitrap Fusion Lumos (Thermo Fisher Scientific). Peptide separation was carried out on an EASY-Spray column (50 cm × 75 μm i.d., PepMap C18, 2-μm

particles, 100 Å pore size, Thermo Fisher Scientific). Mobile phase A consisted of water and 0.1% formic acid. Mobile phase B consisted of 80% acetonitrile and 0.1% formic acid. Peptides were loaded onto the column with 2% B at 300 nL/min flow rate and eluted at 200 nL/min flow rate in two steps: linear increase from 2% B to 40% B for 109 min; then increase from 40% to 95% B for 11 min. The eluted peptides were directly sprayed into the mass spectrometer. Peptides were analysed using a high/high strategy: both MS spectra and MS2 spectra were acquired in the Orbitrap. MS spectra were recorded at a resolution of 120,000. The ions with a precursor charge state between 3+ and 8+ were isolated with a window size of 1.6 m/z and fragmented using high-energy collision dissociation (HCD) with collision energy 30. The fragmentation spectra were recorded in the Orbitrap with a resolution of 30,000. Dynamic exclusion was enabled with a single repeat count and 60 s exclusion duration.

The mass spectrometric raw files were processed into peak lists using ProteoWizard (version 3.0) (Kessner et al, 2008), and crosslinked peptides were matched to spectra using xiSEARCH software (version 1.7.6.1 and 1.7.6.4) (Mendes et al, 2019) (https://github.com/Rappsilber-Laboratory/XiSearch) with preprocessing and in-search assignment of monoisotopic peaks (Lenz et al, 2021). Search parameters were MS accuracy, 3 ppm; MS/MS accuracy, 10 ppm; enzyme, trypsin; crosslinker, BS3 or EDC; max missed cleavages, 4; missing monoisotopic peaks, 2; fixed modification, carbamidomethylation on cysteine; variable modifications, oxidation on methionine and phosphorylation on serine and threonine; fragments, b and y ions with loss of $H_2O$, $NH_3$ and $CH_3SOH$. The mass spectrometry proteomics data have been deposited to the ProteomeXchange Consortium via the PRIDE (Perez-Riverol et al, 2022) partner repository with the dataset identifier PXD056626.

## AlphaFold predictions and identification of CR1-like motifs

AlphaFold prediction was performed using Google Colab multimer (Evans et al, 2022; Jumper et al, 2021; Mirdita et al, 2022) with num_relax set to 5 using protein sequences Sgo1(487–522) or full-length Lrs4, with Ycg1 (6–932 Δ499–555) and Brn1 (384–529). Protein structure visualizations were created with Pymol.

To identify potential Ycg1 interacting proteins we searched for motifs with the consensus [KR]-[KR]-L-[FYVIT]-[KR]-x(1,3)-[IV] in the yeast genome using the Genomenet tool (https://www.genome.jp/tools/motif/) and selected the chromosomal proteins of interest as shown in Fig. EV5G.

### Yeast growth and culture

Cells carrying *pMET-CDC20* were arrested in the G1 phase using alpha factor in synthetic medium lacking methionine for 3 h, then released to rich medium (YPDA) containing methionine for another 2 h to achieve metaphase arrest (tension); or released to YPDA + methionine + nocodazole + benomyl to achieve metaphase arrest (no tension). Alpha factor was used at 5 μg/ml and re-added to 2.5 μg/ml every 1.5 h. Nocodazole was used at 15 μg/ml and re-added to 7.5 μg/ml every 1 h. Benomyl was used at 30 μg/ml. Methionine was used at 8 mM and re-added to 4 mM every 45 min.

### Immunoprecipitation and mass spectrometry

About 1 l of culture grown at $OD_{600} = 1.8$ was treated with 30 μg/ml benomyl for 2 h, and then harvested and drop frozen in liquid nitrogen. Cells were cryo-lysed using a freezer mill (Spex 6875, 8 rounds of 2 min at 10 cycles/second, with 2 min rests). Immuno-precipitation was performed using anti-FLAG-coupled dynabeads as described in "Co-immunoprecipitation using yeast extract". Proteins were eluted from beads in two rounds by incubating with 0.1% RapiGest (Waters) in 50 mM Tris HCl, pH 8.0 at 50 °C for 10 min with 500 rpm shaking.

Protein samples from all biological replicates were processed at the same time and using the same digestion protocol without any deviations. They were subjected to MS analysis under the same conditions. Protein and peptide lists were generated using the same software and the same parameters. Specifically, eluates from each IP were digested using the Filter Aided Sample Preparation (FASP) protocol as described (Wiśniewski et al, 2009) with minor modifications. In brief, IP eluate was reduced with 25 mM DTT at 80 °C for 1 min, then denatured by the addition of urea to 8 M. The sample was applied to a Vivacon 30k MWCO spin filter (Sartorius, UK) and centrifuged at 12.5k × g for 15–20 min. Protein retained on the column was then alkylated with 100 μL of 50 μM iodoacetamide (IAA) in buffer A (8 M urea, 100 mM Tris pH 8.0) in the dark at RT for 20 min. The column was then centrifuged as before, and washed with 100 μL buffer A, then with 2 × 100 μL volumes of 50 mM ammonium bicarbonate (ABC). About 3 μg/μL trypsin (Pierce, UK) in 0.5 mM ABC was applied to the column, which was then incubated at 37° overnight. The eluates from the filter units were acidified using 20 μl of 10% trifluoroacetic acid (TFA) (Sigma Aldrich), and spun onto StageTips as described (Rappsilber et al, 2007). Peptides were eluted in 40 μL of 80% acetonitrile in 0.1% TFA and concentrated down to 1 μL by vacuum centrifugation (Concentrator 5301, Eppendorf, UK). The peptide sample was then prepared for LC-MS/MS analysis by diluting it to 5 μL by 0.1% TFA.

LC-MS analyses were performed on an Orbitrap Exploris™ 480 Mass Spectrometer (Thermo Fisher Scientific, UK) coupled on-line, to an Ultimate 3000 HPLC (Dionex, Thermo Fisher Scientific, UK). Peptides were separated on a 50 cm (2 μm particle size) EASY-Spray column (Thermo Scientific, UK), which was assembled on an EASY-Spray source (Thermo Scientific, UK) and operated constantly at 50 °C. Mobile phase A consisted of 0.1% formic acid in LC-MS grade water, and mobile phase B consisted of 80% acetonitrile and 0.1% formic acid. Peptides were loaded onto the column at a flow rate of 0.3 μL min$^{-1}$ and eluted at a flow rate of 0.25 μL min$^{-1}$ according to the following gradient: 2 to 40% mobile phase B in 150 min and then to 95% in 11 min. Mobile phase B was retained at 95% for 5 min and returned back to 2% a minute until the end of the run (190 min).

Survey scans were recorded at 120,000 resolution (scan range 350–1650 m/z) with an ion target of 5.0e6, and injection time of 20 ms. MS2 data independent acquisition (DIA) was performed in the orbitrap at 30,000 resolution with a scan range of 350–1200 m/z, maximum injection time of 55 ms and AGC target of 3.0E6 ions. We used HCD fragmentation (Olsen et al, 2007) with a stepped collision energy of 25.5, 27, and 30. We used variable isolation windows throughout the scan range ranging from 10.5 to 50.5 m/z. Shorter isolation windows (10.5–18.5 m/z) were applied from 400–800 m/z and then gradually increased to 50.5 m/z until the end of the scan range. The default charge state was set to 3. Data for both survey and MS/MS scans were acquired in profile mode. The DIA-NN software platform (Demichev et al, 2020) version 1.8.1. was used to process the raw files, and a search was conducted

against the *Saccharomyces cerevisiae* complete/reference proteome (Uniprot, released in December, 2019). Precursor ion generation was based on the chosen protein database (automatically generated spectral library) with deep-learning-based spectra, retention time, and IMs prediction. Digestion mode was set to specific with trypsin, allowing a maximum of two missed cleavages. Carbamidomethylation of cysteine was set as a fixed modification. Oxidation of methionine, acetylation of the N-terminus, and phosphorylation of threonine serine and tyrosine were set as variable modifications. The parameters for peptide length range, precursor charge range, precursor m/z range and fragment ion m/z range, as well as other software parameters, were used with their default values. The precursor FDR was set to 1%. Statistics from LFQ data were processed using the Bioconductor *DEP* R package, according to (Zhang et al, 2018). Raw data have been deposited to the ProteomeXchange Consortium via the PRIDE (Perez-Riverol et al, 2022) partner repository with the dataset identifier PXD056626.

### Chromatin immunoprecipitation-qPCR (ChIP-qPCR)

ChIP and qPCR were performed as described previously using anti-HA 12CA5, anti-GFP or anti-FLAG antibody (Su et al, 2021). Primers used for qPCR analysis were 5'-gtacccagcttttgttcccttag-3' and 5'-actttatgcttccggctcctat-3', and the reactions were carried out on a LightCycler 480 machine (Roche) with 45 cycles for NEB Luna. Experiments were done with four repeats and error bars represent standard error.

### Calibrated chromatin immunoprecipitation and sequencing (ChIP-Seq)

ChIP-Seq was done as described previously with minor modifications (Paldi et al, 2020; Barton et al, 2022). Briefly, nocodazole arrested yeast culture was grown and fixed as for ChIP-qPCR. To each condition, an equal amount of *S. pombe* (using AMsp635 for Brn1-6HA ChIP-Seq, and AMsp1863 for Sgo1-FLAG ChIP-Seq) fixed cell pellets were added for calibration. Cells were then processed as for ChIP-qPCR except three rounds of 30 s on Fastprep (BioPulverizer FP120), followed by two rounds of sonication (20 cycles of 30 s on/30 s off, at HIGH setting, Bioruptor Plus Diagenode) were carried out. For every 1 ml IP, 15 μl of Protein G Dynabeads (Invitrogen) and 7.5 μl of 12CA5 anti-HA (Roche; 11583816) or 5 μl of anti-FLAG (Sigma; F1804) antibody were used.

ChIP-sequencing libraries were prepared using NEXTflex-6 DNA Barcodes (PerkinElmer), quantified by Qubit (Thermo Fisher Scientific), and then library quality was tested using the 2100 Bioanalyzer High Sensitivity DNA kit (Agilent, Santa Clara, CA) on Bioanalyzer (Agilent). 1 nM pooled library (INPUT:IP ratio = 15:85%) was sequenced in house on an Illumina MiniSeq instrument (Illumina, San Diego, CA) with MiniSeq High output reagent kit (150-cycles, paired-end) (Illumina).

Data analysis was performed as described previously (Barton et al, 2022). Mean plots (pileup) at 16 centromeres and 32 pericentromeric borders sites were generated using the Deeptools computeMatrix and PlotProfle package. Reads were binned at 50 bp windows around the midpoint of centromeres or borders with 3 or 25 kb flanks at either side. Borders were oriented so that their position relative to the centromere was the same. Centromere and pericentromere border peak coordinates are as described

previously (Paldi et al, 2020). For the rDNA repeats region, reads are randomly assigned to the reference genome containing two copies of rDNA (RDN37-1, RDN5-1 and RDN37-2, RDN5-2).

### Biorientation assay

The biorientation assay was performed as described previously (Verzijlbergen et al, 2014), with minor modifications. Briefly, cells carrying *pMET-CDC20*, *CEN4-GFP*, and *SPC42-tdTomato* were arrested in the G1 phase using alpha factor in a medium lacking methionine for 3 h, then released to rich medium (YPDA) containing methionine, nocodazole, and benomyl. After 3 h treatment, a t0 sample was taken and drugs were washed out and released into rich medium containing methionine to allow spindles to reform while maintaining metaphase arrest. Samples were taken every 20 min by fixing in 3.7% formaldehyde for 10 min and resuspended in 1xDAPI for microscopy. Typically, 200 cells (at least 100 cells) were scored for each timepoint.

### Hi-C

Hi-C was performed as described previously (Paldi et al, 2020). Briefly, cells carrying *pMET-CDC20* were arrested in metaphase with or without tension, and fixed at $OD_{600} = 0.6$ with 3% formaldehyde for 20 min, then quenched with 0.35 M glycine. After washing with cold water, cells were resuspended in NEBuffer 2 and drop frozen in liquid nitrogen. Following grinding, DpnII digestion, biotinylation, ligation, crosslink reversal, and biotin removal, DNA was fragmented by 2-3 rounds of sonication (30 cycles of 30 s on/30 s off, at HIGH setting, Bioruptor Plus Diagenode). After DNA end repair, A-tailing, Ampure XP beads size selection, biotin pull-down and adapter ligation (NEXTflex-6 DNA Barcodes, PerkinElmer), the Hi-C libraries were sequenced at EMBL Core Genomics Facility (Heidelberg, Germany).

For Hi-C data analysis, Fastq reads were aligned to sacCer3 reference genome using Hi-C- Pro v2.11.1 bowtie2 v2.3.4.1 (--very-sensitive -L 30 --score-min L,-0.6,-0.2 --end-to-end --reorder), removing singleton, multi-hit and duplicated reads. Read pairs were assigned to restriction fragment (DpnII), and invalid pairs filtered out. Valid interaction pairs were converted into the cool contact matrix format using the cooler library, and matrixes were balanced using Iterative correction down to one kilobase resolution. Multi-resolution cool files were uploaded onto a local HiGlass server for visualization, cooler show was also used to generate individual plots for each chromosome. The number of valid read pairs for each experiment are given in Table EV3.

To generate pileups at centromeres/pericentromeric borders, the cooltools library was used, and cool matrixes were binned at 1 kb resolutions. Plots were created around the midpoint of centromeres with 25/100 kb flanks on each side, or around the midpoint of borders with 25 kb flanks, showing the $log_{10}$ mean interaction frequency using a color map similar to HiGlass 'fall'. The ratio pileups between samples were created in a similar fashion plotting, the $log_2$ difference between samples in the 'coolwarm' color map, i.e., A/B; red signifying increased contacts in A relative to B and blue decreased contacts in B relative to A.

### Quantification and statistical analysis

Statistical analysis and graphs were generated using GraphPad Prism 10 software (San Diego). Micrographs and graphs were assembled using Adobe Illustrator. Sequence alignment was

generated using JalView (Waterhouse et al, 2009). IP-MS data was analyzed using R studio. AlphaFold model was generated using AlphaFold2. Details of replicates are given in the figure legends.

## Data availability

The datasets and computer code produced in this study are available in the following databases: Proteomics: Immunoprecipitation mass spectrometry datasets reported in this study have been deposited at PRIDE (Perez-Riverol et al, 2022) with the accession number PXD056626 (https://www.ebi.ac.uk/pride/). Proteomics: Crosslinking mass spectrometry datasets reported in this study have been deposited at PRIDE (Perez-Riverol et al, 2022) with the accession number PXD056649. Genomics: Sequencing data for ChIP and Hi-C analysis have been deposited at GEO with the accession number GSE263958 (https://www.ncbi.nlm.nih.gov/geo/query/acc.cgi). Scripts to analyse sequencing data are available at (https://github.com/danrobertson87/).

The source data of this paper are collected in the following database record: biostudies:S-SCDT-10_1038-S44318-024-00336-6.

## Peer review information

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

## Acknowledgements

We gratefully acknowledge the Wellcome Discovery Research Platform for Hidden Cell Biology Proteomics Core and Bioinformatics Core for mass spectrometry and bioinformatics support, respectively. We thank the Edinburgh Protein Purification Facility and Martin Wear for access to equipment. We are grateful to Vladimir Benes and the EMBL Genecore for sequencing Hi-C libraries. We thank Christian Häring and Damien D'Amours for plasmids and yeast strains. We are grateful to Owen Davies, Marcus Hassler, Christian Häring, Daniel Panne, Martin Singleton, and Marcus Wilson for helpful discussions, and to Owen Davies, Alexander Julner Dunn, Lori Koch, Lucia Massari and Hollie Rowlands for comments on the manuscript. We thank Lucia Massari for the template used to generate Fig. 8H. This work was funded through a Wellcome Investigator award to ALM [220780], two Wellcome Multi-User Equipment Grants [108504 and 2183052], core funding for the Wellcome Centre for Cell Biology [203149], and a Wellcome Discovery Research Platform Award [226791].

## Author contributions

**Menglu Wang**: Conceptualization; Data curation; Formal analysis; Investigation; Visualization; Writing—original draft; Writing—review and editing. **Daniel Robertson**: Data curation; Formal analysis. **Juan Zou**: Data curation; Formal analysis. **Christos Spanos**: Data curation; Formal analysis. **Juri Rappsilber**: Supervision. **Adele L Marston**: Conceptualization; Supervision; Funding acquisition; Visualization; Writing—original draft; Writing—review and editing.

Source data underlying figure panels in this paper may have individual authorship assigned. Where available, figure panel/source data authorship is listed in the following database record: biostudies:S-SCDT-10_1038-S44318-024-00336-6.

## Disclosure and competing interests statement

The authors declare no competing interests.

# Expanded View Figures

**Figure EV1.   Analysis of Sgo1-condensin complexes.**

(A) Size exclusion chromatography (SEC) profiles and corresponding silver-stained SDS-PAGE gels and immunoblot using the indicated antibodies for the analysis of full-length Sgo1 (red) and condensin (yellow) complex formation (blue). Note that Sgo1 alone could not readily be detected in this assay as it associates non-specifically with the column in the absence of condensin. The arrowhead indicates the peak of Sgo1-condensin complex. (B) Crosslinking mass spectrometry of full-length Sgo1 with condensin data mapped onto reported condensin cryo-EM structure (PDB 6YVU (Lee et al, 2020)) and part Ycg1-Brn1 crystal structure which is a dimer (PDB 5OQQ (Kschonsak et al, 2017)). Self-links are hidden and only crosslinks with score >10.5 are shown. (C) Purified GFP-tagged recombinant Sgo1 variants co-immunoprecipitated with condensin (Brn1-6HA) from *sgo1Δ* yeast extract (*sgo1Δ* yeast strains: no tag (AM827), Brn1-6HA (AM8834)). Elutes were analysed by immunoblot with the indicated antibodies. Source data are available online for this figure.

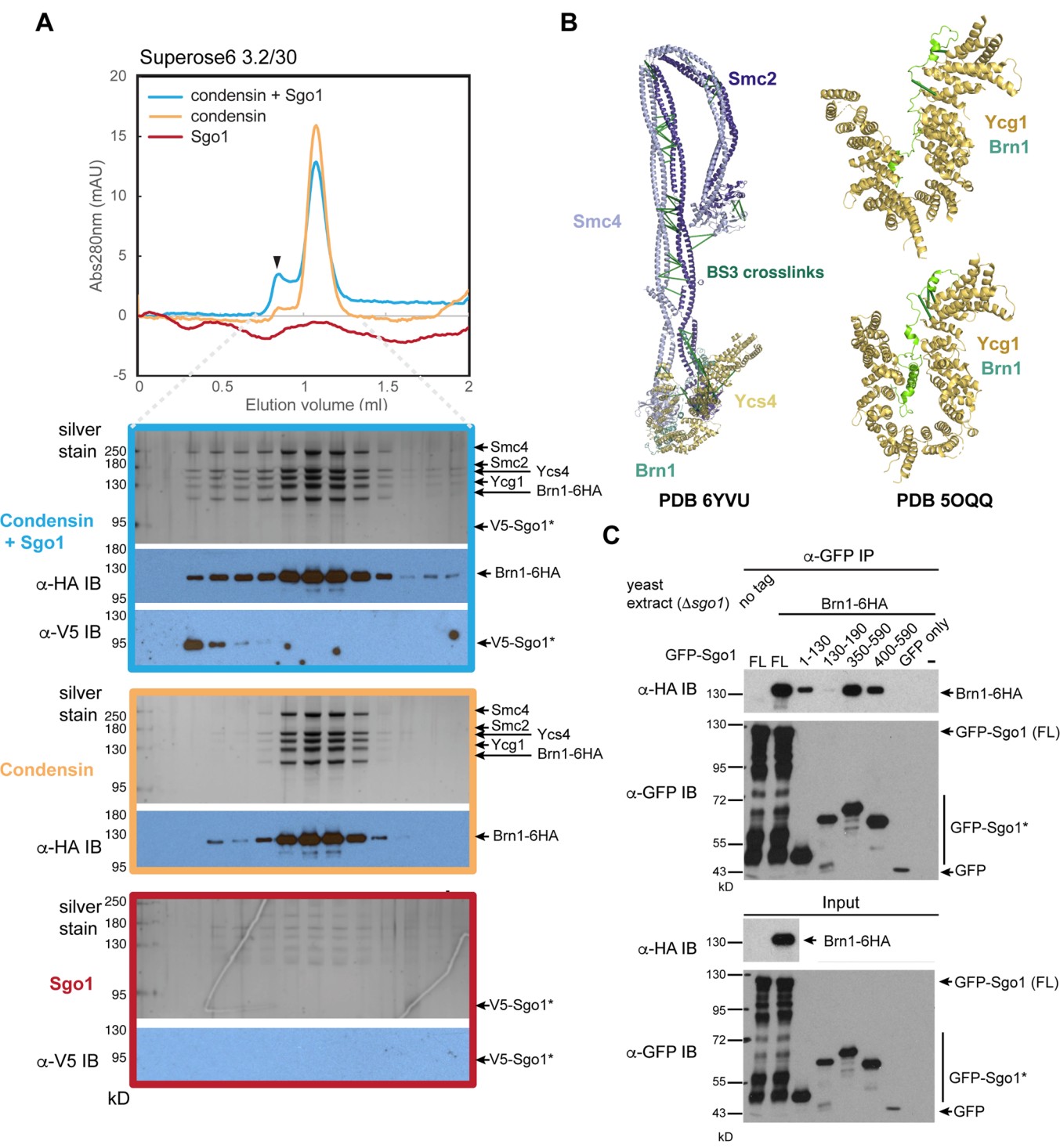

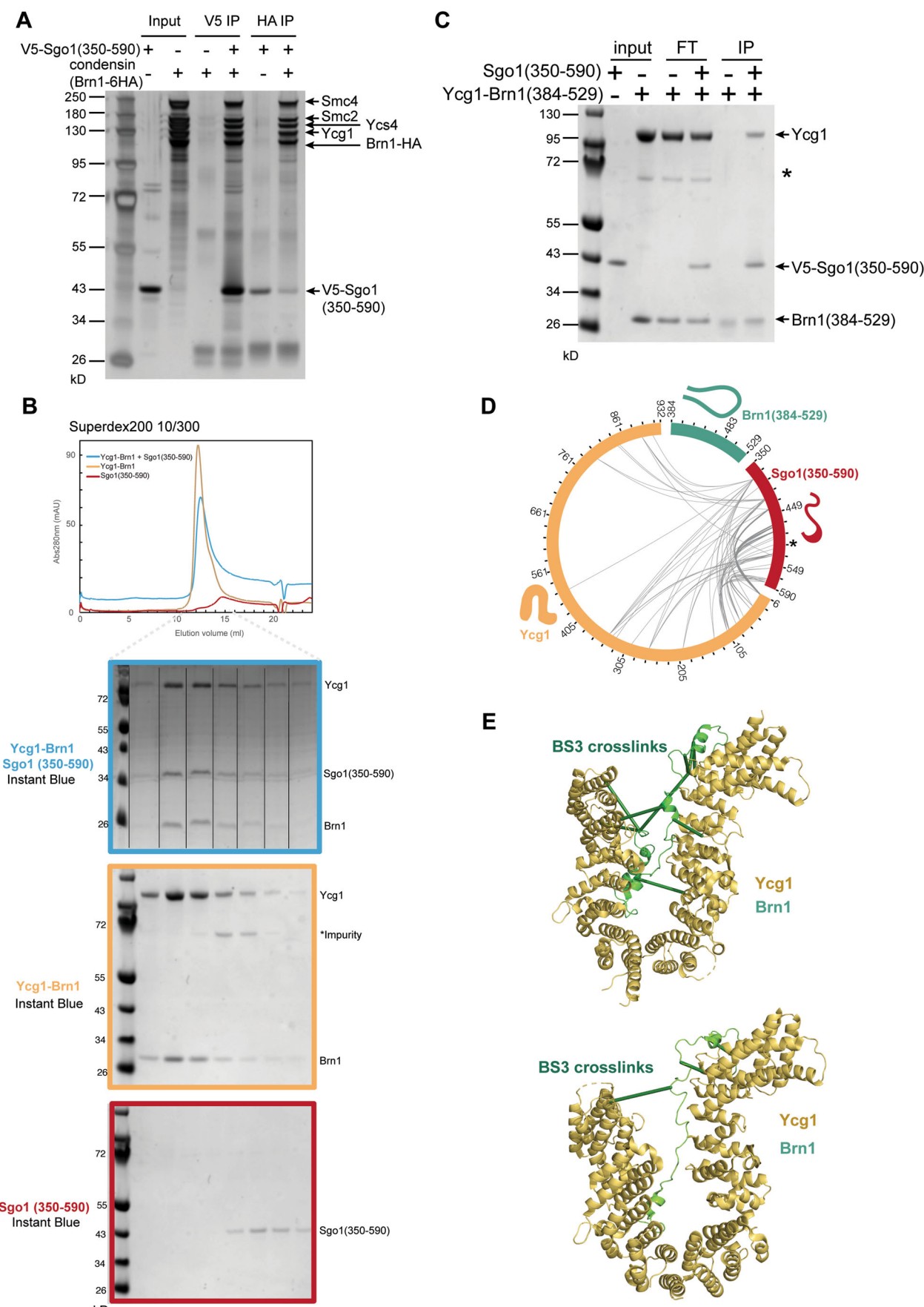

**Figure EV2.  Sgo1 C-terminal region binds directly to Ycg1.**

(A) V5 IP: Recombinant C-terminal region of Sgo (350–590) co-immunoprecipitated with condensin (Brn1-6HA) using V5-coupled beads. HA IP: Recombinant condensin (Brn1-6HA) immunoprecipitated with V5-Sgo1 (350–590). Note Sgo1(350–590) bound non-specifically to anti-HA beads. Eluates were analyzed by silver-stained SDS-PAGE. (B) SEC profiles and corresponding SDS-PAGE for the analysis of interactions (blue) between Sgo1 (350–590) (red) and Ycg1 (6–932, Δ499–555)-Brn1 (384–529) (yellow). Non-relevant lanes have been removed for clarity, as indicated by vertical separation lines. Full gel images are provided in the source data. (C) V5 tagged recombinant C-terminal region of Sgo1 (350–590) co-immunoprecipitated with Ycg1 (6–932, Δ499–555)-Brn1 (384–529). Eluates were analyzed by silver-stained SDS-PAGE. Asterisk indicates impurity, FT is flow through. (D) BS3 crosslinking mass spectrometry of Sgo1 (350–590) and Ycg1 (6–932, Δ499–555)-Brn1 (384–529). The interactions with Brn1 and self-links are hidden. Crosslinks with score >10.5 are chosen. Asterisk highlights Sgo1 residues (L508, F509) predicted to bind Ycg1. (E) Crosslinking mass spectrometry of Sgo1 (350–590) and Ycg1 (6–932, Δ499–555)-Brn1 (384–529) mapped onto Ycg1-short Brn1 crystal structure which is a dimer (PDB 5OQQ (Kschonsak et al, 2017)). Only crosslinks with score >10.5 were shown. Source data are available online for this figure.

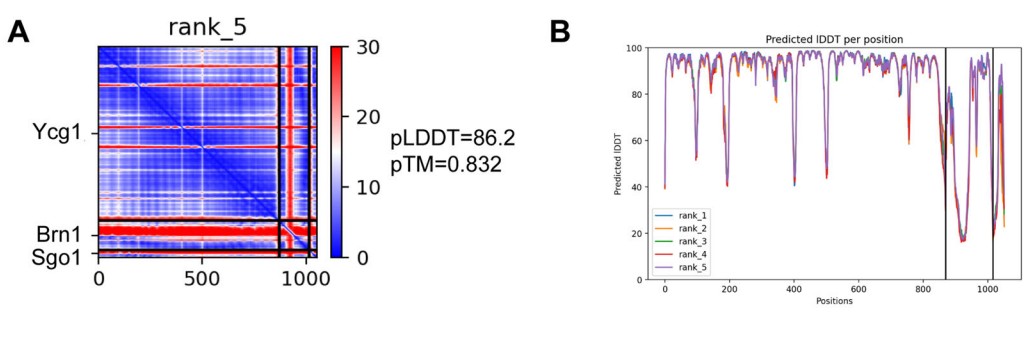

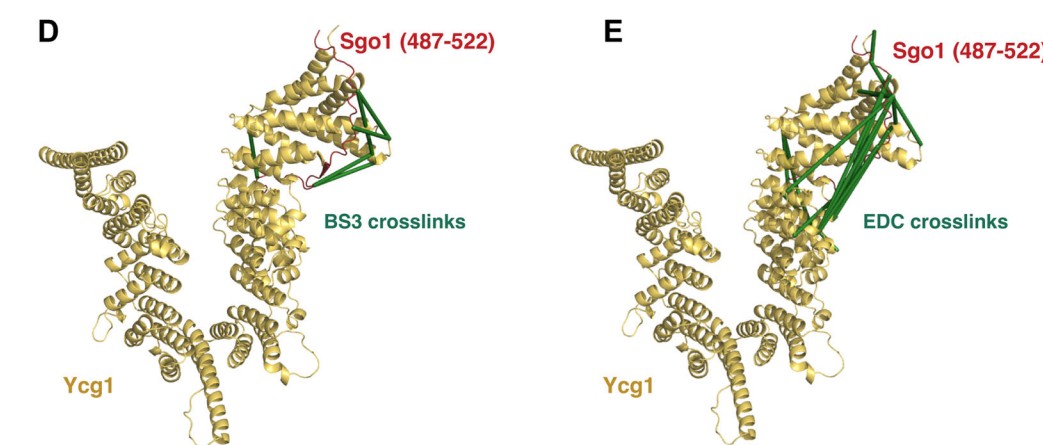

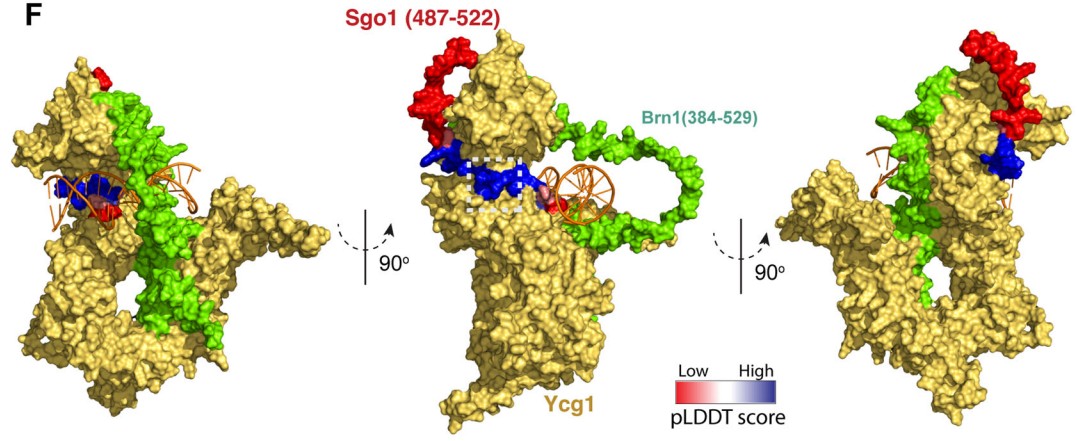

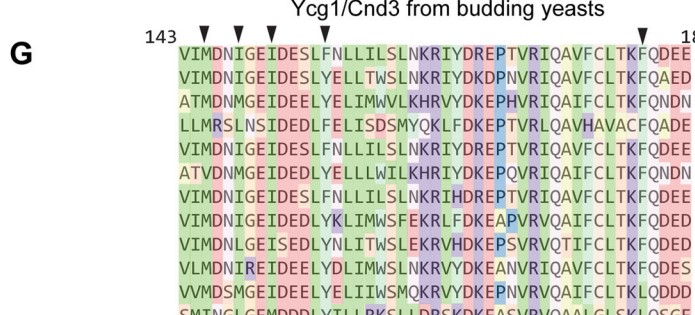

◀ **Figure EV3. Agreement of the AlphaFold model with CLMS data.**

(A) Predicted aligned error (PAE) matrices of AlphaFold2 model_Rank 5 obtained from the Sgo1(487–522) with Ycg1 (6–932, Δ499–555)-Brn1 (384–529) prediction. (B) pLDDT plot of all 5 AlphaFold2 models. (C) Table showing the pLDDT confidence scores of Sgo1 residues L508 and F509 for all five AlphaFold2 models. (D, E) Crosslinking mass spectrometry data of Sgo1 (350–590) and Ycg1 (6–932, Δ499–555)-Brn1 (384–529) with crosslinker BS3 (D) and EDC (E) mapped onto the AlphaFold2 model. (F) DNA was docked onto the AlphaFold2 model by aligning with the reported crystal structure (PDB 5OQP (Kschonsak et al, 2017)). Sgo1 peptide (487–522) is colored with pLDDT score (blue indicates pLDDT >70, red means pLDDT <50). Sgo1 L508, F509 residues are highlighted in the dashed box. (G) Conservation of the Ycg1 binding pocket in related yeast.

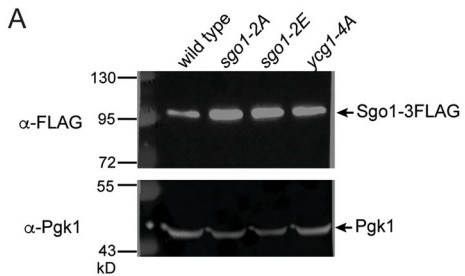

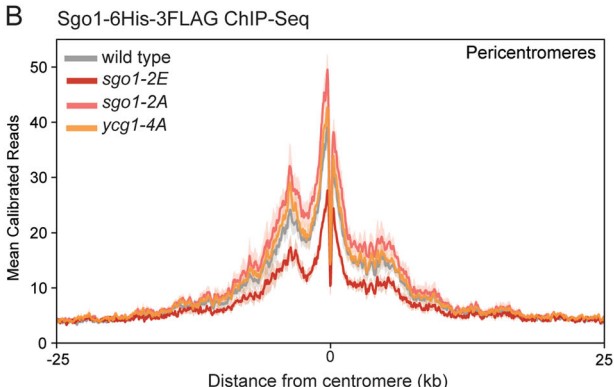

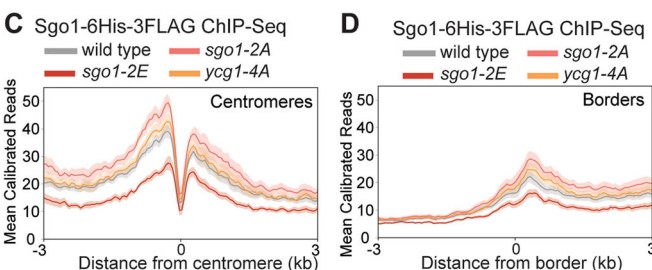

**Figure EV4. Sgo1 enrichment along pericentromeres.**

(A) Anti-FLAG western blotting confirms that Sgo1 is produced at a similar level in all strains. Anti-Pgk1 immunoblot is shown as a loading control. (B–D) Calibrated Sgo1-6His-3FLAG ChIP-Seq using cells arrested in metaphase by treatment with nocodazole. The pileup of pericentromeric region of all 16 chromosomes (B). Zoomed-in pileups of a 6 kb region surrounding 16 centromeres (C) or 32 pericentromeric borders (D). Strains used in calibrated Sgo1-6His-3FLAG ChIP-Seq: *S. cerevisiae:* wild type (AM32740), *sgo1-2E* (AM33140), *sgo1-2A*(AM33145), *ycg1-4A* (AM33265). *S. pombe* used for calibration: AMsp1863. Source data are available online for this figure.

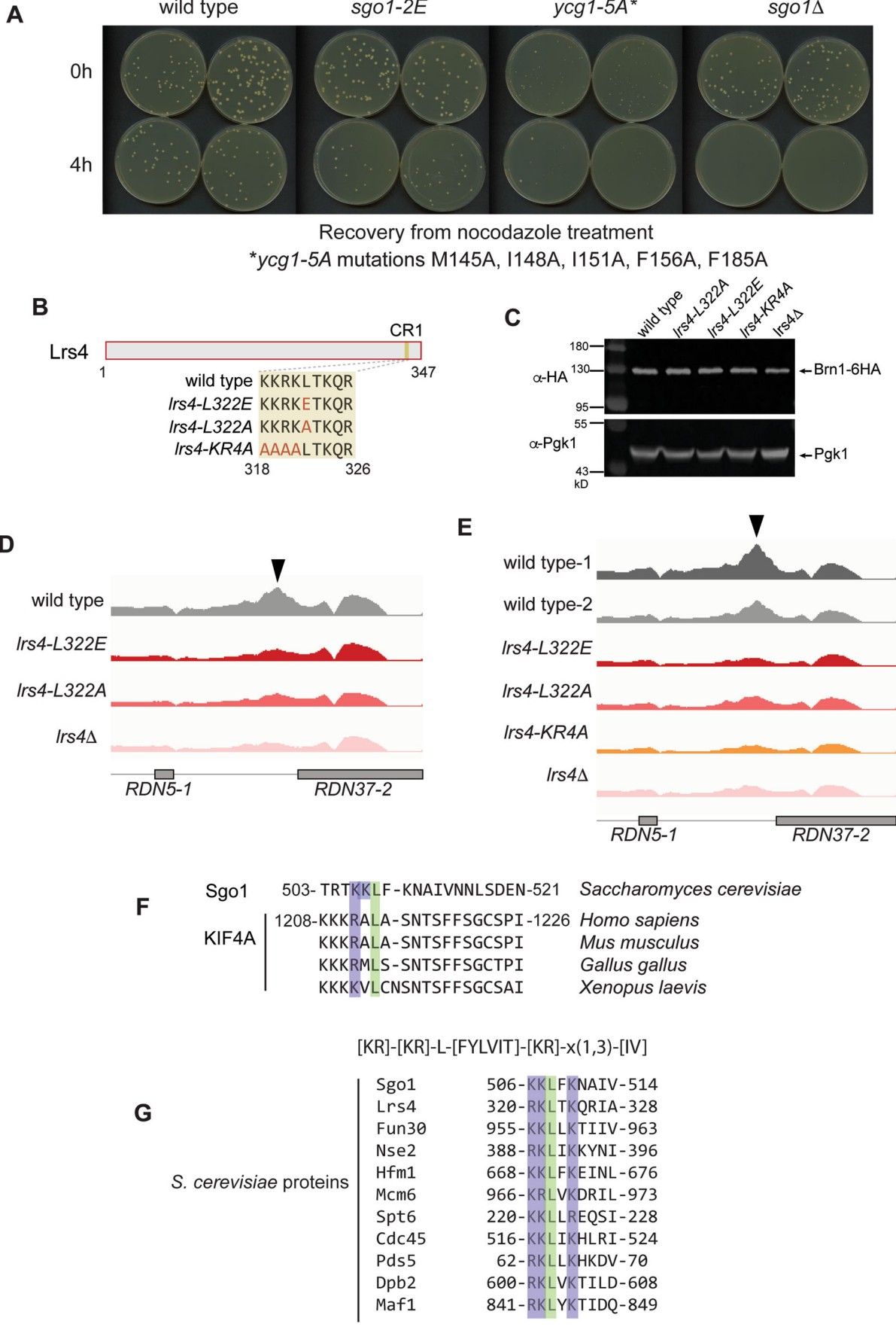

◀

**Figure EV5.   Identification of CR1-like motifs in other potential Ycg1/CAP-G ligands.**

(A) Mutation of the Ycg1 binding pocket reduces colony size to a greater extent than mutation of the Sgo1-CR1. Cells were plated onto a rich medium before (0 h) or after the addition of nocodazole (4 h). Strains used: wild type (AM23137), *sgo1-2E* (AM33044), *ycg1-5A* (AM33315), and *sgo1Δ* (AM827). (B) Scheme showing the endogenous point mutants generated at the CR1 motif of monopolin protein Lrs4. (C) Anti-HA immunoblot showing Brn1 is produced at a similar level in all strains. Anti-Pgk1 immunoblot is shown as a loading control. (D, E) Calibrated condensin (Brn1-6HA) ChIP-Seq of nocodazole arrested cells at rDNA region. Condensin (Brn1-6HA) enrichment peaks are indicated with arrowheads. *S. pombe* strain AMsp635 was used for calibration. (D) FLAG-tagged Lrs4 strains were used: wild type (AM33965), *lrs4-L322E* (AM33967), *lrs4-L322A* (AM33966), *lrs4Δ* (AM9766). (E) Untagged Lrs4 strains were used: wild type-1 (AM5708), wild type-2 (AM34390), *lrs4-L322E* (AM34392), *lrs4-L322A* (AM34391), *lrs4-KR4A* (AM34393), and *lrs4Δ* (AM9766). Note two different wild-type mutants were used to confirm that rDNA phenotypes in point mutant strains were not a consequence of their derivation from an *lrs4Δ* parent. Wild type-2 and all mutants were generated from *lrs4Δ* (AM9766) by standard PCR method; while wild type-1 is a standard wild type. (F) A CR1 motif is found in KIF4A. (G) Potential candidate condensin ligands containing ([KR]-[KR]-L-[FYVIT]-[KR]-x(1,3)-[IV]) in *S. cerevisiae*. Source data are available online for this figure.

