## [Peer Review File · The EMBO Journal]

Molecular mechanism targeting condensin for chromosome condensation

Menglu Wang, Daniel Robertson, Juan Zou, Christos Spanos, Juri Rappsilber, and Adele Marston

Corresponding author(s): Adele Marston (adele.marston@ed.ac.uk)

Review Timeline:

Submission Date:	22nd May 24
Editorial Decision:	9th Jul 24
Revision Received:	10th Oct 24
Editorial Decision:	23rd Nov 24
Revision Received:	26th Nov 24
Accepted:	2nd Dec 24

Editor: Hartmut Vodermaier

Transaction Report:

Prof. Adele L Marston
University of Edinburgh
Wellcome Centre for Cell Biology
Max Born Crescent
Edinburgh EH9 3BF
United Kingdom

9th Jul 2024

Re: EMBOJ-2024-117960
Molecular mechanism targeting chromosome condensation

Dear Adele,

Thank you for submitting your study on condensin-Sgo1 interaction, and apologies for the delayed review process due to referees requiring additional time for back-to-back reviewing. I have now received a complete set of reports, copied below for your information. As you will see, all referees acknowledge the interest and importance of your findings as well as the overall quality of the work, and we would therefore be happy to consider a revised manuscript further for EMBO Journal publication. At the same time, the reviewers do raise several concerns and suggestions, whose incorporation should further strengthen the study.

Since we usually aim for a single round of major revision, I would encourage you to contact me with a tentative point-by-point response/revision plan already during the early stages of your revision work. On the basis of this, we could discuss which points would appear to be the key issues, and how they might best be addressed. As always, our 'scooping protection' will mean that competing work appearing here or elsewhere during the course shall not affect our final considerations of your study.

Further information on preparing, formatting and uploading a revised manuscript can be found below and in our Guide to Authors. Thank you again for the opportunity to consider this work for The EMBO Journal, and I look forward to hearing from you in due time.

With kind regards,

Hartmut

4) Each main and each Expanded View (EV) figure should be uploaded as individual production-quality files (preferably in .eps,

.tif, .jpg formats). For suggestions on figure preparation/layout, please refer to our Figure Preparation Guidelines: <http://bit.ly/EMBOPressFigurePreparationGuideline>

9) To facilitate reproducibility and cross-laboratory adoption of methodologies, please structure the Materials & Methods section as outlined in our guide to authors, including a completed Reagents and Tools Table that can be downloaded from our author guidelines as well (<https://www.embopress.org/page/journal/14602075/authorguide#structuredmethods>).

10) Digital image enhancement is acceptable practice, as long as it accurately represents the original data and conforms to community standards. If a figure has been subjected to significant electronic manipulation, this must be clearly noted in the figure legend and/or the 'Materials and Methods' section. The editors reserve the right to request original versions of figures and the original images that were used to assemble the figure. Finally, we generally encourage uploading of numerical as well as gel/blot image source data; for details see: embopress.org/page/journal/14602075/authorguide#sourcedata

At EMBO Press, we ask authors to provide source data for the main manuscript figures. Our source data coordinator will contact you to discuss which figure panels we would need source data for and will also provide you with helpful tips on how to upload and organize the files.

In the interest of ensuring the conceptual advance provided by the work, we recommend submitting a revision within 3 months (7th Oct 2024). Please discuss the revision progress ahead of this time with the editor if you require more time to complete the revisions. Use the link below to submit your revision:

Link Not Available

Referee #1:

This study set out to dissect the molecular mechanism, as well as the functional significance, of Condensin localization to chromosomes in budding yeast. The authors found that a conserved pocket on the yeast Condensin HAWK subunit Ycg1 interacts with a so-called CR1 motif in Sgo1 and Lrs4 (a yeast homolog of human xxx), which is important for targeting Ycg1 to pericentromeres and rDNA, respectively. Functionally, the authors showed that Sgo1-mediated recruitment of Ycg1 helps shape pericentromeres to favor sister kinetochores bi-orientation, leading to in kinetochore tension that enables the relocalization of Codensin activity to chromosome arms. Overall, these findings are novel and of significance. Conclusions are well supported by the data. Besides, the manuscript is well written. See my major and minor points below.

Major points:

- 1) It would be nice if the authors could repeat the experiments shown in Figure 1B after depleting individual subunit of the Condensin complex, which may show the contribution of each Cohesin subunit to the interaction between the Codensin complex and Sgo1;
- 2) Figure EV1C showed that both Sgo1 N-terminal fragment (1-130) and C-terminal fragments (350-590 and 400-590) were able to pull down Brn1-6xHA. It would be better if the authors could test whether the Sgo1 N-terminal fragment (1-130) also contributes to full-length Sgo1 binds to Condensin subunits, such as Ycg1, in vivo;
- 3) It would be better if authors could explain why deletion of residues 504-519 only partly reduced the interaction of Ycg1 with GFP-Sgo1 (350-590) (In Figure EV1D-1F);

4) It would be nice if authors could discuss why Sgo1 CR1 is conserved only among yeasts closely related to *S. cerevisiae*.

Minor points:

1) In the last paragraph of the Discussion section, the authors describe that "The CES is formed of a composite surface between the HAWK, SA2, and the kleisin RAD21 and is a binding site for multiple regulators^{64,65}. These include the loop anchor protein CTCF, the cohesin release factor, WAPL1 and SGO1, which counteracts WAPL-dependent cohesin release^{64,65}. A recent study reported that the inner kinetochore protein CENP-U also utilizes a F-x-F motif to bind the CES between SA2 and Rad21, as CTCF, Wapl and Sgo1 do (PMID: 38714893), which should be cited.

Referee #2:

Wang et al. uncover how Sgo1 and condensin interact in order to localise condensin to pericentromeres during mitosis. Sgo1 was previously shown to be important for condensin localisation and biorientation, and the authors build on this here to further our understanding of condensin regulation. They identify a conserved pocket on the condensin subunit Ycg1 which interacts with Sgo1. They find that the CR1 motif in Sgo1 is a key player in this interaction and show that in cells, mutations within this interface indeed lead to reduced levels of condensin at pericentromeres and result in defective biorientation. Although the pulldowns show modest changes in binding, the data from cells (MS and ChIP-seq), combined with AlphaFold and the in vitro data is compelling evidence that the described interaction is important to localise condensin. The authors then begin to look at how this might affect the structure of centromeres. However, the effects observed by Hi-C are modest. The conclusions drawn from this latter part of the manuscript are difficult to reconcile with the data, and the corresponding models overall are really difficult to follow.

In my view, the strength of this manuscript lies in the identification of the binding site on Ycg1 and its interaction with Sgo1. The authors also present exciting evidence that there are other ligands which may bind to the same site on Ycg1 to recruit condensin to different chromosomal regions. With this in mind, my suggestion would be to rewrite this manuscript in such a manner that the strongest data (the binding interface and condensin recruitment) are the main focus. This part is not only interesting and of value to the field, but it is also of very high quality, and is suitable for publication as is. In the manuscript's current guise, the genome architecture part (i.e. the Hi-C experiments) gets too much attention, and includes several unclear and weak models. This is unnecessary. The Hi-C could still be a feature as a minor part of the manuscript, with fewer models based on the relatively weak evidence. With the conclusions drawn from the Hi-C being less bold and more in line with the minor observations, and with the focus shifted towards the binding interface, I believe that this could become an excellent manuscript whose publication I would strongly support.

Specific comments:

- Title, abstract and discussion: The claims made here are too bold. The authors indeed show the mechanism behind condensin recruitment to pericentromeres and provide a compelling argument that Ycg1 is recruited via the same conserved patch to rDNA, but they do not directly show that this is through Lrs4. I also do not feel that there is sufficient evidence to confirm that condensin shapes pericentromeres, as the effects are modest. The models based on this bit of data are not well supported by the evidence.
- There are a few minor typos which should be corrected throughout the manuscript.
- EV1B: It is odd that a complex lacking Ycg1 is used here. It would make more sense to also show Ycg1 in proximity to SMC4 as referred to in the next sentence. Particularly as the rest of the manuscript focusses on Ycg1.
- EV1C: The levels of the different Sgo1 fragments can't be assessed in the blots shown due to the high exposure and background. Please show a lower exposure to allow for this.
- EV2D: There appear to be many crosslinks before residue 500. Please clarify in the text.
- EV3B-F: The authors conclude that CR1 is the major Ycg1 binding site. However, the results indicate that CR2 is also involved in binding to condensin. In EV3F, the short CR1 deletion (504-519) is similar to the short CR2 deletion (559-573). It is only when a larger CR1 deletion (487-522) is made that significant changes in binding becomes apparent. If a larger deletion in CR2 was also made, would there also be a larger effect on binding? I agree that CR1 appears to be a prominent binding site but the authors do not rule out that CR2 is also significantly binding. This should be made clear in the conclusions drawn from this part.
- Fig 1G: The L508E/F509E mutations show a significant difference but the L508E/F509E in combination with other mutations expected to disrupt binding do not. How is this possible? Do these additional mutations rescue the binding?
- EV6B-C: Sgo1-2A appears to affect all PP2A subunits, and shows a lower intensity for all cohesin subunits when compared to wild type. Ycg1-4A also appears to consistently affect the cohesin subunits to some extent. Although condensin is clearly affected the most, the authors cannot conclude that these mutations do not affect the other complexes at all.
- Fig 5-6: The Hi-C plots are difficult to understand. In part A, the negative scale is not intuitive. If this is to be used, more details in the figure legend are needed. The scale/colours should also be better described in the legend, particularly for part B/C. The tension/no tension labels would be more clear if used to directly label the axis rather than in the box above the plot. It would also be helpful if the authors would add labels/arrows to point out what is being described in the text.
- Fig 5C: The ChIP-seq has no labels, is it a Brn1 ChIP-seq as in previous figures?
- Fig 5 E/F: Are the mutants A or E? The figures and the text do not match.

- Fig 5: The authors conclude that the structural differences must be due to condensin. It is clear that in these mutants the overwhelming phenotype is that condensin fails to localise to pericentromeres (Fig 4). However, in figure 5 Ycg1-4A has a very mild defect and Sgo1-2A even less so. The defect is only really clear in Sgo1 deficient cells. This suggests that these structural changes are perhaps not simply due to condensin, and that other roles of Sgo1 may also be involved. One potential explanation could be, as mentioned above, that the mutants have a minor effect on PP2A/cohesin. There is therefore not sufficient evidence to conclude that there are condensin-dependent structural changes at the centromeres under tension. From the plots shown, the effects of condensin on these structures is minor. Further work would be needed to draw such a conclusion. For example: if condensin is depleted or knocked out, are these structural changes also apparent and to the same extent as the mutants?
- Fig 5G: The proposed model is not clear and is difficult to reconcile with the presented Hi-C data.
- Fig 6: The authors write that the Sgo1-2A mutant is like WT. However, there are clearly differences relative to WT, and it appears to be the opposite of other mutants in that there are more mid-range contacts in the mutant under tension (Fig 6C). Again, the observations from the Hi-C data are modest and the proposed model is difficult to understand in this context. Just like in the panels mentioned above, the Sgo1 deficient cells show the clearest structural changes, suggesting that this may not be explained by condensin alone. But with such minor differences it is quite hard to make any conclusions.
- Fig 6D: A zoom-in of the key contacts would help to visualise the data. It is currently difficult to see the differences between the conditions.
- Fig 7 E/F: The text on the scale bars is too small and unreadable.
- The authors should indicate in the methods which software was used to create protein structure visualisations.

Referee #3:

Condensin plays important roles in the folding of mitotic chromosomes. In this manuscript, Wang et al. identify a conserved patch of the HEAT repeat subunit of condensin (Ycg1 in yeast) that is critical for its recruitment to pericentromeres and rDNA. They show that a small motif termed CR1 in Sgo1 and Lrs4, interacts with this patch in Ycg1. They provide compelling evidence to show that Sgo1 CR1 is required for the recruitment of condensin to pericentromeres and for proper chromosome bi-orientation. The study provides key mechanistic insight into condensin regulation in yeast cells and should be published.

I have the following suggestions that the authors may wish to consider in revising their manuscript.

Major points

- (1) The title and the abstract have overstated their conclusions. There are other mechanisms that regulate chromosome condensation. In addition, even though the Ycg1 patch is important for chromosome arm condensation, the Sgo1 CR1 motif is not. The authors have speculated that this discrepancy might be due to the presence of a yet unidentified condensin receptor at chromosome arms. This might be true, but without the identification of this receptor, it would be premature to claim that "Here, we define the molecular mechanism targeting the condensin SMC complex to chromosomes and reveal how this directs ordered, region-specific chromosome folding.". These conclusions need to be toned down.
- (2) In Fig. 7, the authors demonstrate the importance of the Ycg1 patch in targeting condensin to rDNA and in rDNA condensation. They did not show that the Lrs4 motif is important. They may wish to add experiments testing the potential importance of the Lrs4 motif.

Referee #1:

This study set out to dissect the molecular mechanism, as well as the functional significance, of Condensin localization to chromosomes in budding yeast. The authors found that a conserved pocket on the yeast Condensin HAWK subunit Ycg1 interacts with a so-called CR1 motif in Sgo1 and Lrs4 (a yeast homolog of human xxx), which is important for targeting Ycg1 to pericentromeres and rDNA, respectively. Functionally, the authors showed that Sgo1-mediated recruitment of Ycg1 helps shape pericentromeres to favor sister kinetochores bi-orientation, leading to kinetochore tension that enables the relocalization of Condensin activity to chromosome arms. Overall, these findings are novel and of significance. Conclusions are well supported by the data. Besides, the manuscript is well written. See my major and minor points below.

We thank the reviewer for their comments and support of our manuscript.

Major points:

1) It would be nice if the authors could repeat the experiments shown in Figure 1B after depleting individual subunit of the Condensin complex, which may show the contribution of each Cohesin subunit to the interaction between the Condensin complex and Sgo1;

We provide substantial evidence that the key contacts between condensin and Sgo1 are via the Ycg1 subunit: (a) recombinant Sgo1(350-590) interacts with recombinant Ycg1-Brn1, (b) cross-linking mass spectrometry indicates that the major cross-links are between Sgo1 and Ycg1, (c) Specific mutations at the Sgo1-Ycg1 interface predicted by alphafold are sufficient to abrogate the interaction *in vivo*. Testing the requirement for each individual subunit as the reviewer suggests would involve generating new constructs, multiple new protein purifications and binding assays. These are challenging experiments due to the difficulty in purifications and would take several months. Although the question raised by the reviewer is interesting, we believe that the effort and time involved outweighs the new insight to be gained from these experiments.

2) Figure EV1C showed that both Sgo1 N-terminal fragment (1-130) and C-terminal fragments (350-590 and 400-590) were able to pull down Brn1-6xHA. It would be better if the authors could test whether the Sgo1 N-terminal fragment (1-130) also contributes to full-length Sgo1 binds to Condensin subunits, such as Ycg1, *in vivo*;

We agree with the reviewer that it would be interesting to further investigate the contribution of the N terminal region of Sgo1 in condensin binding. However, this region of Sgo1 also binds PP2A, and potentially CPC-Aurora B. Due to these confounding factors and the substantial amount of work involved in dissecting out the mechanism, we decided to focus our study on the C-terminal part of Sgo1. Our findings show that abrogation of the CR1 motif was sufficient to abolish the interaction with condensin *in vivo*, so even if the N terminus contributes, it is not sufficient *in vivo*. Molecular analysis of the N-terminal portion of Sgo1 will be the subject of a future study in our group.

3) It would be better if authors could explain why deletion of residues 504-519 only partly reduced the interaction of Ycg1 with GFP-Sgo1 (350-590) (In Figure EV1D-1F);

We believe the reviewer is referring to the data that was shown in Fig. EV3D-F and is now in Fig. 2C-E. The *in vitro* studies with recombinant Sgo1 were challenging because Sgo1, and its fragments, are sticky and bind non-specifically not only to beads but also to other proteins. The reviewer makes a good point as in this assay GFP-Sgo1(350-590) lacking residues 504-519 retains some ability to bind Ycg1, while deletion also of the CR2 motif further reduced binding. In principle, this could be due to loss of either specific or non-specific binding. Although from this assay alone it is not possible to distinguish between these possibilities, taking all of the *in vitro* and *in vivo* data into account suggests that only the CR1 interaction is specific, while enhancement of binding due to the other parts of the protein is non-specific. First, in the *in vitro* assay (now in Figure 2C and D), the general observation was that with larger size deletions, the greater the reduction in binding. However, the exception to this is deletion of the CR1 (504-519) or CR2 (559-573) motif alone, where deletion of CR1 has a greater effect than deletion of CR2. Second, in Figure 2A and B we show that fragments containing the CR1 motif tend to bind to Ycg1 better than fragments without it. Third, our *in vivo* data both from tethering (now Figure S1) and IP-MS (now Figure 4 and Figure S2) demonstrate that residues within CR1 are essential for the Sgo1 interaction with condensin. We added the following sentence to this section to explain this: "Although Sgo1 tended to bind beads non-specifically, direct comparisons among a panel of recombinant fragments focused our attention on the CR1 motif as the most critical determinant for binding."

4) It would be nice if authors could discuss why Sgo1 CR1 is conserved only among yeasts closely related to *S. cerevisiae*.

This is a good suggestion. We added a short paragraph to the end of the section of the discussion entitled "Specific targeting of SMC proteins to pericentromeres"

Minor points:

1) In the last paragraph of the Discussion section, the authors describe that "The CES is formed of a composite surface between the HAWK, SA2, and the kleisin RAD21 and is a binding site for multiple regulators^{64,65}. These include the loop anchor protein CTCF, the cohesin release factor, WAPL1 and SGO1, which counteracts WAPL-dependent cohesin release^{64,65}. A recent study reported that the inner kinetochore protein CENP-U also utilizes a F-x-F motif to bind the CES between SA2 and Rad21, as CTCF, Wapl and Sgo1 do (PMID: 38714893), which should be cited.

This is a good suggestion, thank you for pointing this out, we apologise for not including this relevant and important study in the first version of our manuscript and have now referred to this work in this section of the discussion.

Referee #2:

Wang et al. uncover how Sgo1 and condensin interact in order to localise condensin to pericentromeres during mitosis. Sgo1 was previously shown to be important for condensin localisation and biorientation, and the authors build on this here to further our understanding of condensin regulation. They identify a conserved pocket on the condensin subunit Ycg1 which interacts with Sgo1. They find that the CR1 motif in Sgo1 is a key player in this interaction and show that in cells, mutations within this interface indeed lead to reduced levels of condensin at pericentromeres and result in defective biorientation. Although the pulldowns show modest changes in binding, the data from cells (MS and ChIP-seq), combined with AlphaFold and the in vitro data is compelling evidence that the described interaction is important to localise condensin. The authors then begin to look at how this might affect the structure of centromeres. However, the effects observed by Hi-C are modest. The conclusions drawn from this latter part of the manuscript are difficult to reconcile with the data, and the corresponding models overall are really difficult to follow.

In my view, the strength of this manuscript lies in the identification of the binding site on Ycg1 and its interaction with Sgo1. The authors also present exciting evidence that there are other ligands which may bind to the same site on Ycg1 to recruit condensin to different chromosomal regions. With this in mind, my suggestion would be to rewrite this manuscript in such a manner that the strongest data (the binding interface and condensin recruitment) are the main focus. This part is not only interesting and of value to the field, but it is also of very high quality, and is suitable for publication as is. In the manuscript's current guise, the genome architecture part (i.e. the Hi-C experiments) gets too much attention, and includes several unclear and weak models. This is unnecessary. The Hi-C could still be a feature as a minor part of the manuscript, with fewer models based on the relatively weak evidence. With the conclusions drawn from the Hi-C being less bold and more in line with the minor observations, and with the focus shifted towards the binding interface, I believe that this could become an excellent manuscript whose publication I would strongly support.

We thank the reviewer for their careful reading of our manuscript and support of our study. We agree with the reviewer that the genome architecture part raises several questions, that some parts need further explanation and that some of the conclusions should be toned down. However, we believe that inclusion of the Hi-C in our manuscript is important as it adds functional context to the specific localisation of condensin. Therefore, we prefer to keep this data in our manuscript but have appropriately revised our conclusions from this part of the study and removed most of the models derived from it.

Specific comments:

- Title, abstract and discussion: The claims made here are too bold. The authors indeed show the mechanism behind condensin recruitment to pericentromeres and provide a compelling argument that Ycg1 is recruited via the same conserved patch to rDNA, but they do not directly show that this is through Lrs4. I also do not feel that there is sufficient evidence to confirm that condensin shapes pericentromeres, as the effects are modest. The models based on this bit of data are not well supported by the evidence.

We have now generated the specific mutations in Lrs4 and performed condensin ChIP-Seq, as requested by reviewer 3. This shows that recruitment of condensin to the rDNA is indeed through the CR1 on Lrs4. This data is now shown in EV Fig. 5B-E and included in results section "A common condensin recruitment mechanism: Ycg1 conserved patch and Lrs4 CR1 motif recruit condensin to the rDNA". Note that this ChIP-Seq experiment had higher background than our previous ChIP-Seq, which we believe is a problem with the batch of HA antibody used. However, we made two separate sets of yeast strains (one with a FLAG tag on Lrs4, one without) and in both cases, the condensin peak was reduced to a similar level to the *lrs4Δ*, which was previously shown to lose condensin in the rDNA. In light of these results we believe that the title and the statement in the abstract referring to Lrs4 and the rDNA are justified.

We did, however re-write the abstract to focus on the discovery of the CR1 motif and its functional importance in Sgo1 and Lrs4. We toned down the statements surrounding the effects on chromosome arm condensation and the potential existence of further CR1-containing proteins.

We have also re-written the discussion to avoid over-speculation related to chromosome condensation. We removed the section "Licensing chromosome arm condensation upon sister kinetochore biorientation" and instead integrated the following text into the first discussion section "Interestingly, we found evidence that inter-sister kinetochore tension in metaphase also initiates condensation of chromosome arms. This requires Sgo1, but not its binding to Ycg1. The CPC is important for chromosome condensation^{8,62,63} and understanding the interplay between Sgo1, CPC and condensin in biorientation and chromosome condensation is an important priority."

We also removed the more speculative models from the figures,

- There are a few minor typos which should be corrected throughout the manuscript.

We will carefully proofread the manuscript and corrected the typos.

- EV1B: It is odd that a complex lacking Ycg1 is used here. It would make more sense to also show Ycg1 in proximity to SMC4 as referred to in the next sentence. Particularly as the rest of the manuscript focusses on Ycg1.

There are no high-resolution structures of condensin including Ycg1. This figure is just a control to show that the complex used in the cross-linking mass spectrometry adopts the expected confirmation. We have now mapped crosslinks onto a structure of Ycg1-Brn1 and included this in EV Figure 1 to confirm that our crosslinks in Ycg1 are also valid.

- EV1C: The levels of the different Sgo1 fragments can't be assessed in the blots shown due to the high exposure and background. Please show a lower exposure to allow for this.

We have shown a lower exposure, as requested.

- EV2D: There appear to be many crosslinks before residue 500. Please clarify in the text.

We now refer to residues after 470. The exact positions for the crosslinking data are also given in the supplementary tables.

- EV3B-F: The authors conclude that CR1 is the major Ycg1 binding site. However, the results indicate that CR2 is also involved in binding to condensin. In EV3F, the short CR1 deletion (504-519) is similar to the short CR2 deletion (559-573). It is only when a larger CR1 deletion (487-522) is made that significant changes in binding becomes apparent. If a larger deletion in CR2 was also made, would there also be a larger effect on binding? I agree that CR1 appears to be a prominent binding site but the authors do not rule out that CR2 is also significantly binding. This should be made clear in the conclusions drawn from this part.

Note this is now Figure 2A-E. As described in the response to reviewer 1 as copied below, we believe this is due to non-specific binding which is a feature of Sgo1.

The in vitro studies with recombinant Sgo1 were challenging because Sgo1, and its fragments, are sticky and bind non-specifically not only to beads but also to other proteins. The reviewer makes a good point as in this assay GFP-Sgo1(350-590) lacking residues 504-519 retains some ability to bind Ycg1, while deletion also of the CR2 motif further reduced binding. In principle, this could be due to loss of either specific or non-specific binding. Although from this assay alone it is not possible to distinguish between these possibilities, taking all of the in vitro and in vivo data into account, suggests that only the CR1 interaction is specific, while enhancement of binding due to the other parts of the protein is non-specific. First, in the in vitro assay (Figure 2C-E), the general observation was that with larger size deletions, the greater the reduction in binding. However, the exception to this is deletion of the CR1 (504-519) or CR2 (559-573) motif alone, where deletion of CR1 has a greater effect than deletion of CR2. Second, in Figure 2A and B we show that fragments containing the CR1 motif tend to bind to Ycg1 better than fragments without it. Third, our in vivo data both from tethering (Appendix Figure S1) and IP-MS (Figure 4 and Appendix Figure S2) demonstrate that residues within CR1 are essential for the Sgo1 interaction with condensin. We added the following sentence to this section to explain this: Although Sgo1 tended to bind beads non-specifically, direct comparisons among a panel of recombinant fragments focused our attention on the CR1 motif as the most critical determinant for binding."

- Fig 1G: The L508E/F509E mutations show a significant difference but the L508E/F509E in combination with other mutations expected to disrupt binding do not. How is this possible? Do these additional mutations rescue the binding?

This is now Figure 2H. Thank you to the reviewer for pointing this out. As the reviewer indicates, it appears that additional mutations rescue the binding of L508E/F509E, at least in this in vitro assay, and we agree that this is curious and unexpected. Due to the caveats of the in vitro assay and the high level of variability (as clear from

the individual data points) as explained above, we decided not to pursue this, particularly as similar mutations do not rescue binding in the tethering assay (now Figure S1). We added the following sentence: “We note that, unexpectedly, combining Sgo1 I513E, V514E and L517E mutations with L508E F509E increased Ycg1 binding *in vitro* (Fig. 2G, H), but this was not the case in the *in vivo* tethering assay with full length protein (Fig. S1B), suggesting the additional glutamic acid substitutions increase non-specific binding of Sgo1 fragments *in vitro*.”

- EV6B-C: Sgo1-2A appears to affect all PP2A subunits, and shows a lower intensity for all cohesin subunits when compared to wild type. Ycg1-4A also appears to consistently affect the cohesin subunits to some extent. Although condensin is clearly affected the most, the authors cannot conclude that these mutations do not affect the other complexes at all.

We apologise for the mis-understanding due to providing insufficient detail to interpret the graphs which are now shown in Appendix Fig. S2. As the reviewer points out, the amount of Sgo1-2A pulled down is lower than the wild type in this experiment, likely due to technical reasons and accordingly, the amount of PP2A and cohesin co-immunoprecipitating is also slightly lower. The graphs show the values scaled to the mean of all conditions, so they show relative rather than absolute comparisons. Whether any differences are significant is better learned from the volcano plot. The volcano plot comparing wild type and *sgo1-2A* shows clearly that PP2A and cohesin subunits do not significantly change in abundance in the *sgo1-2A* mutant as the points corresponding to all subunits are below the dotted lines defining significance. We have provided a better explanation of the plots in Fig. 4B-E and Appendix Fig. S2 in the figure legend to help the reader understand. We have also labelled PP2A subunits on the volcano so that it is clear that there is no significant difference.

- Fig 5-6: The Hi-C plots are difficult to understand. In part A, the negative scale is not intuitive. If this is to be used, more details in the figure legend are needed. The scale/colours should also be better described in the legend, particularly for part B/C. The tension/no tension labels would be more clear if used to directly label the axis rather than in the box above the plot. It would also be helpful if the authors would add labels/arrows to point out what is being described in the text.

Thank you for this helpful feedback. We have made the suggested changes. This is now Figure 6-7.

- Fig 5C: The ChIP-seq has no labels, is it a Brn1 ChIP-seq as in previous figures?

Thank you for pointing this out. Yes, it is a Brn1 ChIP-Seq. We added labels and description in the figure legend.

- Fig 5 E/F: Are the mutants A or E? The figures and the text do not match.

Apologies for this mistake. It is Ycg1-4A, we fixed this. This is now Figure 6J/K.

- Fig 5: The authors conclude that the structural differences must be due to condensin. It is clear that in these mutants the overwhelming phenotype is that condensin fails to localise to pericentromeres (Fig 4). However, in figure 5 Ycg1-4A has a very mild defect and Sgo1-2A even less so. The defect is only really clear in Sgo1 deficient cells. This suggests that these structural changes are perhaps not simply due to condensin, and that other roles of Sgo1 may also be involved. One potential explanation could be, as mentioned above, that the mutants have a minor effect on PP2A/cohesin. There is therefore not sufficient evidence to conclude that there are condensin-dependent structural changes at the centromeres under tension. From the plots shown, the effects of condensin on these structures is minor. Further work would be needed to draw such a conclusion. For example: if condensin is depleted or knocked out, are these structural changes also apparent and to the same extent as the mutants?

Our prior work and that from the Storchova lab showed that Ipl1 (Aurora B) was not maintained at centromeres in *sgo1* delete cells which results in an error correction phenotype (incorrect attachments cannot be destabilised). We also showed in that publication that condensin mutants have a milder biorientation phenotype than *sgo1* delete cells and that condensin mutants can perform error correction. So the stronger phenotype of the *sgo1* delete is expected as they lose both condensin and error correction. We thank the reviewer for highlighting this issue which we should have explained more clearly. We have re-written this section to make these points clear and focus on the presented results without over-speculation.

- Fig 5G: The proposed model is not clear and is difficult to reconcile with the presented Hi-C data.

We have simplified the model to simply describe the observations without over-speculation. This is now shown in Fig. 6L. Note that the structure of the wild type pericentromere and its dependence on cohesin was determined in our previous publication (Paldi, Nature 2020). The revised model in Fig. 6L shows the effects of impaired biorientation on pericentromere structure in the *ycg1-4A/sgo1-2A/sgo1-2E/sgoD* cells.

- Fig 6: The authors write that the Sgo1-2A mutant is like WT. However, there are clearly differences relative to WT, and it appears to be the opposite of other mutants in that there are more mid-range contacts in the mutant

under tension (Fig 6C). Again, the observations from the Hi-C data are modest and the proposed model is difficult to understand in this context. Just like in the panels mentioned above, the Sgo1 deficient cells show the clearest structural changes, suggesting that this may not be explained by condensin alone. But with such minor differences it is quite hard to make any conclusions.

This is now Figure 7. We agree that the *sgo1* delete shows the clearest structural changes and that this is apparent also from the P(s) curve. Although there could be a very slight increase in mid-range contacts under tension in the *Sgo1-2A* mutant, the evidence to support this is not strong. First, the light blue colour which the reviewer interprets as an increase in mid-range contacts is very close to zero on the scale, meaning no change between wild type and mutant (Figure 7B). Second, the P(s) curve does not indicate an increase in contacts under tension and the difference between the "no tension" curve is very similar to wild type. We therefore changed the wording in the text to state "Surprisingly, however, *sgo1-2A* cells were proficient in chromosome arm condensation under tension".

- Fig 6D: A zoom-in of the key contacts would help to visualise the data. It is currently difficult to see the differences between the conditions.

We have added ovals to identify the relevant part of the plot in the heatmaps and an arrowhead to the contact probability curve. Note this is now Figure 7A-D.

- Fig 7 E/F: The text on the scale bars is too small and unreadable.

We will correct this. Note this is now Figure 8E-G.

- The authors should indicate in the methods which software was used to create protein structure visualisations.

It was Pymol and we added this information to the methods.

Referee #3:

Condensin plays important roles in the folding of mitotic chromosomes. In this manuscript, Wang et al. identify a conserved patch of the HEAT repeat subunit of condensin (Ycg1 in yeast) that is critical for its recruitment to pericentromeres and rDNA. They show that a small motif termed CR1 in Sgo1 and Lrs4, interacts with this patch in Ycg1. They provide compelling evidence to show that Sgo1 CR1 is required for the recruitment of condensin to pericentromeres and for proper chromosome bi-orientation. The study provides key mechanistic insight into condensin regulation in yeast cells and should be published.

We are grateful to the reviewer for their strong support of their manuscript.

I have the following suggestions that the authors may wish to consider in revising their manuscript.

Major points

(1) The title and the abstract have overstated their conclusions. There are other mechanisms that regulate chromosome condensation. In addition, even though the Ycg1 patch is important for chromosome arm condensation, the Sgo1 CR1 motif is not. The authors have speculated that this discrepancy might be due to the presence of a yet unidentified condensin receptor at chromosome arms. This might be true, but without the identification of this receptor, it would be premature to claim that "Here, we define the molecular mechanism targeting the condensin SMC complex to chromosomes and reveal how this directs ordered, region-specific chromosome folding.". These conclusions need to be toned down.

We agree with this assessment. As described in the response to reviewer 2 we have revised the abstract and relevant sections in the text to tone down our conclusions.

(2) In Fig. 7, the authors demonstrate the importance of the Ycg1 patch in targeting condensin to rDNA and in rDNA condensation. They did not show that the Lrs4 motif is important. They may wish to add experiments testing the potential importance of the Lrs4 motif.

Thanks for this great suggestion. We carried out this experiment and confirmed that the CR1 motif in Lrs4 is important for condensin recruitment to the rDNA. This new data is shown in Fig. EV5.

Prof. Adele L Marston
University of Edinburgh
Wellcome Centre for Cell Biology
Max Born Crescent
Edinburgh EH9 3BF
United Kingdom

23rd Nov 2024

Re: EMBOJ-2024-117960R
Molecular mechanism targeting chromosome condensation

Dear Adele,

Thank you for submitting your revised manuscript to The EMBO Journal. Two of the original referees have now assessed it once again (see comments below), and both of them are overall satisfied and only request minor textual revisions. Following their incorporation, as well as addressing of a few remaining editorial issues as follows, we shall therefore be happy to accept the study for publication:

- Figure EV2B seems to contain an undeclared splice between the marker lane and the rest of the gel. Could you therefore please send us (and upload with the other Source Data) annotated raw images for the three "Instant Blue" panels? Should there have been removal of non-relevant lanes (as I suspect), this nevertheless needs to be explicitly mentioned in the respective figure legend, and clearly indicated by separation lines in the gel images.
- Please note that the source data files for the EV figures should be uploaded combined into one single ZIP archive (while for the main figure SD, the current single ZIP file/folder per figure is correct).
- As we are switching from a free-text author contribution statement towards a more formal statement based on Contributor Role Taxonomy (CRediT) terms, please remove the present Author Contribution section and instead specify each author's contribution(s) directly in the Author Information page of our submission system during upload of the final manuscript. See <https://casrai.org/credit/> for more information.
- Please provide suggestions for a short 'blurb' text prefacing and summing up the study in two sentences (max. 250 characters), followed by 3-5 one-sentence 'bullet points' with brief factual statements of key results of the paper; they will form the basis of an editor-written 'Synopsis' accompanying the online version of the article. Please also upload a synopsis image, which can be used as a "visual title" for the synopsis section of your paper. The image should be in PNG or JPG format with the modest dimensions of EXACTLY 550 pixels wide and 300-600 pixels high (maybe based on a condensed version of Figure 8H?).

I am therefore returning the study to you once more, to allow you to incorporate these final changes and upload all requested files. Once we have received them, we should be able to swiftly proceed with acceptance and production of the manuscript.

With kind regards,

Hartmut

- 1) Every manuscript requires a Data Availability section (even if only stating that no deposited datasets are included). Primary datasets or computer code produced in the current study have to be deposited in appropriate public repositories prior to resubmission, and reviewer access details provided in case that public access is not yet allowed. Further information: embopress.org/page/journal/14602075/authorguide#dataavailability
- 2) Each figure legend must specify

9) To facilitate reproducibility and cross-laboratory adoption of methodologies, please structure the Materials & Methods section as outlined in our guide to authors, including a completed Reagents and Tools Table that can be downloaded from our author guidelines as well (<https://www.embopress.org/page/journal/14602075/authorguide#structuredmethods>).

10) Digital image enhancement is acceptable practice, as long as it accurately represents the original data and conforms to community standards. If a figure has been subjected to significant electronic manipulation, this must be clearly noted in the figure legend and/or the 'Materials and Methods' section. The editors reserve the right to request original versions of figures and the original images that were used to assemble the figure. Finally, we generally encourage uploading of numerical as well as gel/blot image source data; for details see: embopress.org/page/journal/14602075/authorguide#sourcedata

At EMBO Press, we ask authors to provide source data for the main manuscript figures. Our source data coordinator will contact you to discuss which figure panels we would need source data for and will also provide you with helpful tips on how to upload and organize the files.

In the interest of ensuring the conceptual advance provided by the work, we recommend submitting a revision within 3 months (21st Feb 2025). Please discuss the revision progress ahead of this time with the editor if you require more time to complete the revisions. Use the link below to submit your revision:

Link Not Available

Referee #1:

In this manuscript, the Marston lab addresses the molecular mechanism targeting budding yeast condensing to specific chromosome regions, which is a key question in chromosome biology field. They find that Sgo1 utilizes a motif called CR1 to bind a conserved pocket on the condensing HAWK subunit Ycg1, and that this interaction recruits condensing to pericentromeres to promote sister kinetochore biorientation. They additionally show that Lrs4 has a similar CR1 motif, which binds Ycg1 and recruits Ycg1 to the rDNA region, leading to rDNA condensation. Overall, this study is well-designed and

appropriately performed. The results are properly interpreted. The data support the conclusions and have implications for future studies to identify potential CR1-containing proteins that may interact with condensins in higher eukaryotes. The manuscript is well-written, for which I really enjoyed reading. I only have several minor suggestions for the authors to consider.

1. It would be helpful if authors could discuss why the predicted Sgo1 CR1-binding site on Ycg1 is highly conserved even in mammals, whereas the Sgo1 CR1 is conserved only among yeasts closely related to budding yeast.
2. The authors showed in Fig. EV4A-D that Sgo1-2A and ycg1-4A mutations do not greatly perturb the cellular levels or localization of Sgo1 itself, whereas Sgo1-2E showed a general reduction in its association with chromatin for reasons that are currently unclear. It would be nice if the authors could discuss why they used the Sgo1-2E mutant, rather than the Sgo1-2A mutant, to demonstrate the Sgo1-dependent recruitment of Condensin to pericentromeres facilitates sister kinetochore biorientation in mitosis.
3. In the Discussion, the authors mention that "Finally, Sgo1 engages condensin to promote proper sister kinetochore biorientation. Sgo1 also ensures biorientation by localizing the Chromosome Passenger Complex (CPC) containing Aurora B at kinetochores, which destabilising tension-less kinetochore microtubule attachments to provide a further opportunity for correct attachments to be made (Verzijlbergen et al, 2014; Peplowska et al, 2014)", I would suggest to cite three highly relevant papers (PMID: 32028528; PMID: 32027339; PMID: 31868888).
4. Given that both CENP-U (Yan et al, 2024) and Sgo1 (García-Nieto et al, 2023) counteract WAPL-dependent cohesin release from centromeres García-Nieto et al, 2023; Yan et al, 2024, the sentence in the Discussion "These include the loop anchor protein CTCF, the kinetochore protein CENP-U, the cohesin release factor, WAPL1, and SGO1, which counteracts WAPL-dependent cohesin release (Li et al, 2020; García-Nieto et al, 2023; Yan et al, 2024)" would be better to change to "These include the loop anchor protein CTCF, the cohesin release factor Wapl, as well as Sgo1 and the kinetochore protein CENP-U which counteract Wapl-dependent cohesin release (Li et al, 2020; García-Nieto et al, 2023; Yan et al, 2024)".

Referee #2:

The authors have satisfactorily addressed my comments. I now consider this nice paper ready for publication!

Editorial comments

- Figure EV2B seems to contain an undeclared splice between the marker lane and the rest of the gel. Could you therefore please send us (and upload with the other Source Data) annotated raw images for the three "Instant Blue" panels? Should there have been removal of non-relevant lanes (as I suspect), this nevertheless needs to be explicitly mentioned in the respective figure legend, and clearly indicated by separation lines in the gel images.

Apologies that we missed this. We have added vertical lines to the figure, updated the figure legend and provided the source files as requested.

- Please note that the source data files for the EV figures should be uploaded combined into one single ZIP archive (while for the main figure SD, the current single ZIP file/folder per figure is correct).

We have now provided this in the correct format.

- As we are switching from a free-text author contribution statement towards a more formal statement based on Contributor Role Taxonomy (CRediT) terms, please remove the present Author Contribution section and instead specify each author's contribution(s) directly in the Author Information page of our submission system during upload of the final manuscript. See <https://casrai.org/credit/> for more information.

We removed the text from the manuscript as requested.

- Please provide suggestions for a short 'blurb' text prefacing and summing up the study in two sentences (max. 250 characters), followed by 3-5 one-sentence 'bullet points' with brief factual statements of key results of the paper; they will form the basis of an editor-written 'Synopsis' accompanying the online version of the article. Please also upload a synopsis image, which can be used as a "visual title" for the synopsis section of your paper. The image should be in PNG or JPG format with the modest dimensions of EXACTLY 550 pixels wide and 300-600 pixels high (maybe based on a condensed version of Figure 8H?).

We provided these as requested.

Referee #1:

In this manuscript, the Marston lab addresses the molecular mechanism targeting budding yeast condensing to specific chromosome regions, which is a key question in chromosome biology field. They find that Sgo1 utilizes a motif called CR1 to bind a conserved pocket on the condensing HAWK subunit Ycg1, and that this interaction recruits condensing to pericentromeres to promote sister kinetochore biorientation. They additionally show that Lrs4 has a similar CR1 motif, which binds Ycg1 and recruits Ycg1 to the rDNA region, leading to rDNA condensation. Overall, this study is well-designed and appropriately performed. The results are properly interpreted. The data support the conclusions and have implications for future studies to identify potential CR1-containing proteins that may interact with condensins in higher

eukaryotes. The manuscript is well-written, for which I really enjoyed reading. I only have several minor suggestions for the authors to consider.

We are grateful to the reviewer for their support and helpful comments.

1. It would be helpful if authors could discuss why the predicted Sgo1 CR1-binding site on Ycg1 is highly conserved even in mammals, whereas the Sgo1 CR1 is conserved only among yeasts closely related to budding yeast.

We had already addressed this in our revised manuscript with addition of this text in the discussion (Page 14) "Whether shugoshins also play a role in condensin recruitment to pericentromeres in other organisms is not known, however we found no obvious CR1 motif in vertebrate shugoshins. Potentially other CR1-containing proteins concentrate condensins in pericentromeres in vertebrates. Alternatively, vertebrate shugoshins may harbour a cryptic CR1 motif that is not easily recognisable by sequence homology."

2. The authors showed in Fig. EV4A-D that Sgo1-2A and ycg1-4A mutations do not greatly perturb the cellular levels or localization of Sgo1 itself, whereas Sgo1-2E showed a general reduction in its association with chromatin for reasons that are currently unclear. It would be nice if the authors could discuss why they used the Sgo1-2E mutant, rather than the Sgo1-2A mutant, to demonstrate the Sgo1-dependent recruitment of Condensin to pericentromeres facilitates sister kinetochore biorientation in mitosis.

We planned to make both mutations. However, for technical, rather than biological reasons, generating the sgo1-2A mutant was delayed. Since the proteomics showed that condensin binding was perturbed to the same extent in both mutations, we decided to go ahead with the sgo1-2E mutant and not wait for the sgo1-2A mutant. Since this is merely a case of us not generating the strain in a timely fashion we decided not to mention this in the manuscript.

3. In the Discussion, the authors mention that "Finally, Sgo1 engages condensin to promote proper sister kinetochore biorientation. Sgo1 also ensures biorientation by localizing the Chromosome Passenger Complex (CPC) containing Aurora B at kinetochores, which destabilising tension-less kinetochore microtubule attachments to provide a further opportunity for correct attachments to be made (Verzijlbergen et al, 2014; Peplowska et al, 2014)", I would suggest to cite three highly relevant papers (PMID: 32028528; PMID: 32027339; PMID: 31868888).

We included these references as suggested by the reviewer.

4. Given that both CENP-U (Yan et al, 2024) and Sgo1 (García-Nieto et al, 2023) counteract WAPL-dependent cohesin release from centromeres García-Nieto et al, 2023; Yan et al, 2024, the sentence in the Discussion "These include the loop anchor protein CTCF, the kinetochore protein CENP-U, the cohesin release factor, WAPL1, and SGO1, which counteracts WAPL-dependent cohesin release (Li et al, 2020; García-Nieto et al, 2023; Yan et al, 2024)" would be better to change to "These

include the loop anchor protein CTCF, the cohesin release factor Wapl, as well as Sgo1 and the kinetochore protein CENP-U which counteract Wapl-dependent cohesin release (Li *et al*, 2020; García-Nieto *et al*, 2023; Yan *et al*, 2024)".

CTCF also counteracts WAPL-dependent release. We have therefore changed this sentence to: "These include the loop anchor protein CTCF, the kinetochore protein CENP-U and SGO1, all of which compete with the cohesin release factor WAPL that also binds to this site (Li *et al*, 2020; García-Nieto *et al*, 2023; Yan *et al*, 2024)."

Referee #2:

The authors have satisfactorily addressed my comments. I now consider this nice paper ready for publication!

We are grateful to the reviewer for their support and helpful comments.

Prof. Adele L Marston
University of Edinburgh
Wellcome Centre for Cell Biology
Max Born Crescent
Edinburgh EH9 3BF
United Kingdom

2nd Dec 2024

Re: EMBOJ-2024-117960R1
Molecular mechanism targeting condensin for chromosome condensation

Dear Prof. Marston,

Thank you for submitting your final revised manuscript for our consideration. I am pleased to inform you that we have now accepted it for publication in The EMBO Journal.

Yours sincerely,

Hartmut Vodermaier
